# Predictability of Non-Phase-Locked Baroclinic Tides in the Caribbean Sea

Edward D. Zaron[1]

[1]Department of Civil and Environmental Engineering, Portland State University, Portland, Oregon, USA

**Correspondence:** E. D. Zaron (ezaron@pdx.edu)

**Abstract.** The predictability of the sea surface height expression of baroclinic tides is examined with 96 hr forecasts produced by the AMSEAS operational forecast model during 2013–2014. The phase-locked tide, both barotropic and baroclinic, is identified by harmonic analysis of the 2 year record and found to agree well with observations from tide gauges and satellite altimetry within the Caribbean Sea. The non-phase-locked baroclinic tide, which is created by time-variable mesoscale stratification and currents, may be identified from residual sea level anomalies (SLAs) near the tidal frequencies. The predictability of the non-phase-locked tide is assessed by measuring the difference between a forecast – centered at $T + 36$ hr, $T + 60$ hr, or $T + 84$ hr – and the model's later verifying analysis for the same time. Within the Caribbean Sea, where a baroclinic tidal sea level range of $\pm 5$ cm is typical, the forecast error for the non-phase-locked tidal SLA is correlated with the forecast error for the sub-tidal (mesoscale) SLA. Root-mean-square values of the former range from $0.5$ cm to $2$ cm, while the latter ranges from $1$ cm to $6$ cm, for a typical $84$ hr forecast. The spatial and temporal variability of the forecast error is related to the dynamical origins of the non-phase-locked tide and is briefly surveyed within the model.

## 1 Introduction

Sea level fluctuations of several centimeters associated with the astronomically-forced baroclinic tide are nearly ubiquitous throughout the ocean (Ray and Mitchum, 1996; Zhao et al., 2016). While they are a relatively small component of the sea level variability spectrum, they can be the dominant source of variability for wavelengths between, roughly, 100 km and 180 km, particularly near their sources (Ray and Zaron, 2011; Zaron, 2017). This component of sea level variability is also associated with subsurface isopycnal variability and baroclinic currents. Baroclinic tides are sometimes regarded as a source of high-frequency noise in ocean observations; however, they are of interest in their own right because of the momentum, energy, and material transports associated with these waves.

It is of interest to know the degree to which baroclinic tidal sea level variability can be predicted. The long record of observations from satellite altimeters has enabled the identification and mapping of the baroclinic sea level phase-locked with the astronomical tidal forcing. This component of sea level is predictable from the orbital elements of the sun and moon, and it is found to closely obey the theoretically-predicted dispersion relation for linear waves propagating through the ocean's time-mean stratification (Dushaw, 2002; Zhao et al., 2011; Ray and Zaron, 2016). In fact, it is only possible to separate the baroclinic tide from the barotropic tide in altimetry data because of the large separation in spatial scales between these classes

of waves (Zaron, 2019). However, there is another component of sea level variability associated with the tidal frequencies that represents non-phase-locked baroclinic tides, which are created by temporal modulations of the propagation medium (Munk and Cartwright, 1966; Rainville and Pinkel, 2006; Colosi and Munk, 2006; Zilberman et al., 2011; Ray and Zaron, 2011). Because modulations of the propagation medium – caused by mesoscale eddies and other processes – are, in part, represented within operational ocean forecasting systems, it ought to be possible to predict some component of the non-phase-locked tide with such a forecasting system.

This paper investigates the predictability of sea level associated with the non-phase-locked baroclinic tide in the AMSEAS model, a state-of-the-art operational ocean forecasting system. The emphasis on sea level was chosen (rather than, say, ocean currents) because of its relation to studies of ocean surface topography with satellite altimeters. But even this narrow emphasis poses challenges because of the wide spectrum of non-tidal variability present in sea level. Thus, this study utilizes steric height (which is derived from the forecast model outputs), rather than total sea level, in order to separate the baroclinic tidal signals from broadband high-frequency barotropic variability related to winds and atmospheric pressure. Another challenge for this study is identification of non-phase-locked baroclinic tidal signals in observational data. Neither tide gauge data, nor altimeter data, permit a unique identification of these signals; and model-data comparisons of high-frequency variability are limited by the AMSEAS output products, which are only available at three hour intervals. For these reasons, the question of predictability is assessed using self-verifying analyses, where the model output on a subsequent date is used to validate forecasts generated on previous dates. This measure of forecast error thus provides a best-case estimate or lower-bound on the forecast skill to be expected with independent data.

Preparation for the Surface Water & Ocean Topography (SWOT) swath altimeter mission, planned for launch in 2021, is concerned with distinguishing balanced motion from inertia-gravity waves in sea surface topography data (Zaron and Rocha, 2018), and the sea surface expression of baroclinic tides is a prominent manifestation of inertia-gravity waves. The present study provides a baseline measure of model forecast skill which will be useful for evaluating future improvements in forecasting baroclinic tides resulting from either changes in the numerical model, the assimilation methodology, or the ocean observation system. The results quantify the forecast skill and provide insight into the phenomenology of internal tide signals as represented in high-resolution ocean models.

## 2  The AMSEAS ocean forecasting system

The AMSEAS model is a 1/30-degree (approx. 3.5 km), 40-level, implementation of the Navy Coastal Ocean Model (NCOM; Kara et al., 2006) which has been producing operational forecasts of the Caribbean Sea, Gulf of Mexico, and Western Atlantic since May 2010. The name, "AMSEAS," is not an acronym, it is the captialized contraction of "American Seas" which has been adopted as the system's name. The model is re-initialized daily by assimilating observations using the Navy Coupled Ocean Data Assimilation System (NCODA; Cummings, 2011) and integrated to produce a 96-hour forecast by the Naval Oceanographic Office (NAVOCEANO). Prior to April 2013, AMSEAS was forced by wind stress and heat flux from the Fleet Numerical Meteorology and Oceanography Center's Navy Operational Global Atmospheric Prediction System (NOGAPS;

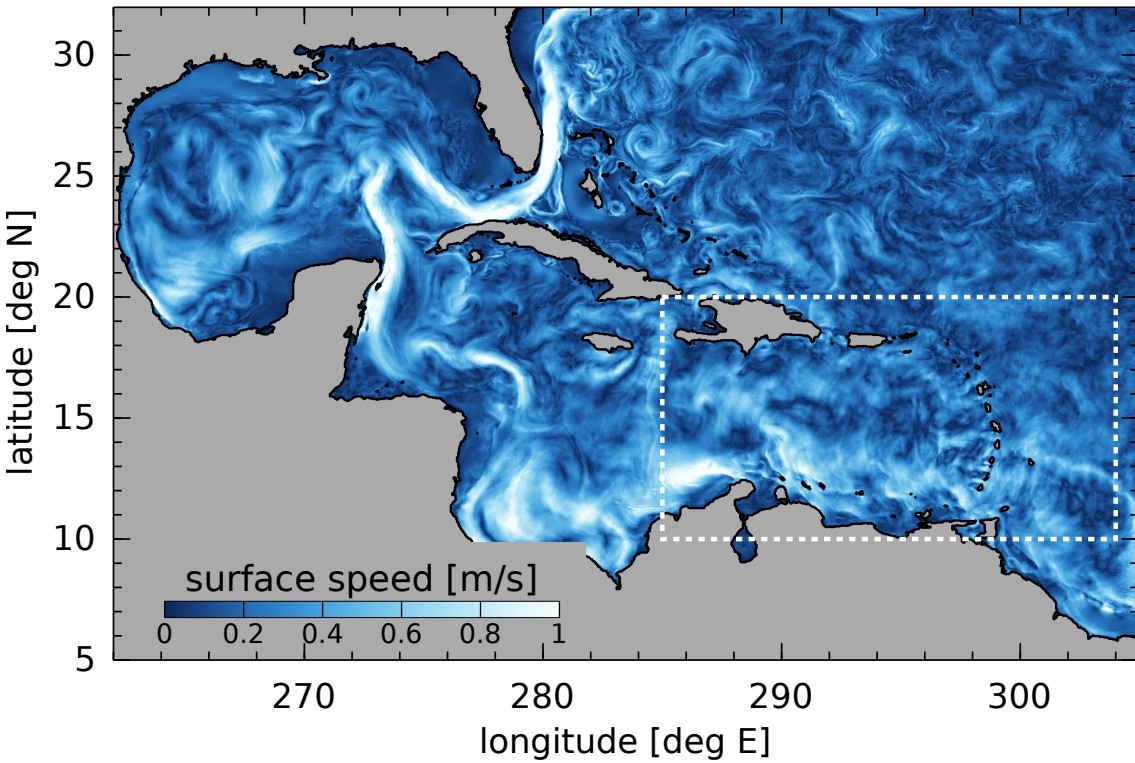

**Figure 1.** AMSEAS model domain. This snapshot of surface currents from the representative date, 2013-01-02, shows prominent features of ocean circulation in the Western Atlantic, namely the Loop Current in the Gulf of Mexico and the Florida Current. The rectangular box (dashed white line) indicates the region shown in subsequent plots. (Here and throughout this manuscript dates are expressed in the ISO 8601 format, year-month-day.)

Rosmond et al., 2002), and lateral open boundary conditions for temperature, salinity, and non-tidal surface elevation were provided by operational Global NCOM (Barron et al., 2007). Since April 2013, the atmospheric forcing has been provided by the Navy Global Environmental Model (NAVGEM; Hogan et al., 2014), with lateral boundary conditions provided by operational Global HYCOM (Metzger et al., 2014). Tides are not included in the HYCOM presently used for boundary conditions.

5   To incorporate them in AMSEAS, the tides predicted with the OTIS barotropic tide model (Egbert and Erofeeva, 2002) are added to the barotropic currents and sea surface heights data at open boundaries. In addition, a tide-generating force is applied within the model domain which is a combination of astronomical forcing, ocean loading, and ocean self-attraction, consistent with the OTIS model.

    The spatial domain of AMSEAS covers the Gulf of Mexico, Caribbean Sea, and a portion of the northeast Atlantic Ocean

10   (Figure 1). The major current systems such as the Yucatan Current, Loop Current, and Florida Current are represented in the model, as well as a broad spectrum of variability related to mesoscale eddies, wind-driven circulation, and tides, consistent with historical observations (Carton and Chao, 1999; Centurioni and Niiler, 2003; Torres and Tsimplis, 2011).

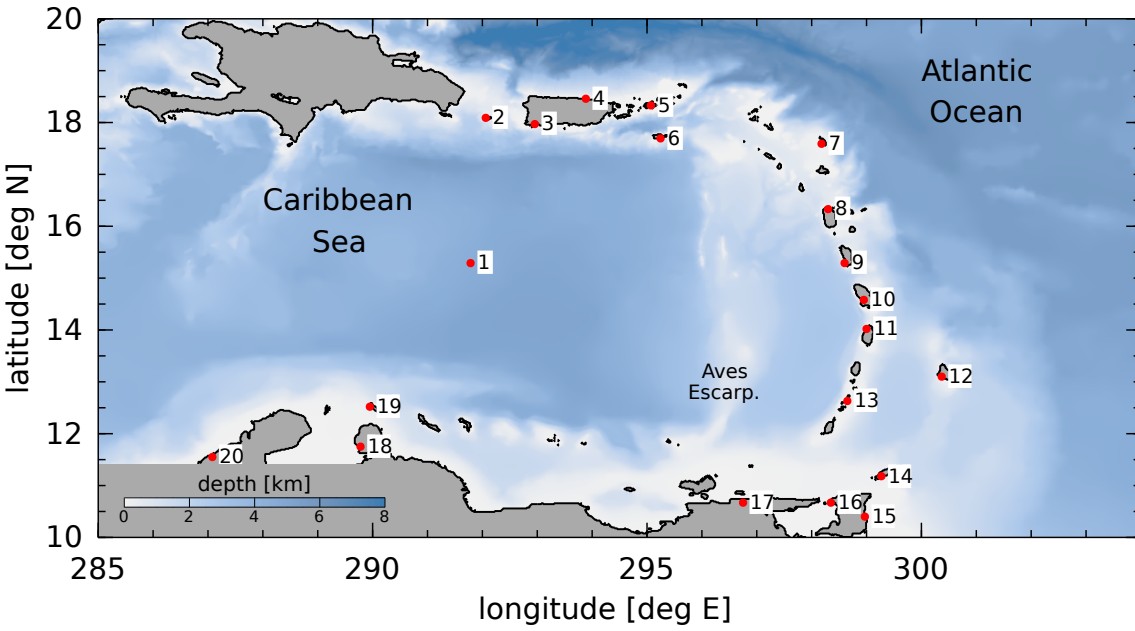

**Figure 2.** Caribbean Sea and tide gauge locations. Tide gauge stations are numbered clockwise around the Caribbean Sea, starting at station 1, DART-42407 (see Table A1). Some closely-spaced stations are not plotted. Significant sites of internal tide generation are Mona Passage (near station 2), Anagada Passage (east of stations 5 and 6), and the passages between the southern Windward Islands (stations 9 to 13) and Grenada Passage (between stations 13 and 17).

AMSEAS nowcast/forecast products have been used and validated in a number of studies. For example, Lagrangian trajectories and forecasts were used to interpret biological observations in the Gulf of Mexico (Nero et al., 2013; O'Conner et al., 2016), and skill assessment efforts in the Gulf of Mexico have been reported (Hernandez et al., 2015; Zaron et al., 2015). AMSEAS was used to provide boundary conditions for a high-resolution regional forecasting system around Puerto Rico and the U.S. Virgin Islands, and validated through comparisons with tide gauge and water current measurements (Solano et al., 2018).

For later reference it is useful to refer to Figure 2 which shows the bottom topography in the region of study. A major topographic feature of the Eastern Caribbean, Aves Escarpment, is indicated, as are the locations of 17 tide gauge sites, including one bottom pressure gauge (site 1). Evidence for the range of dynamics active in AMSEAS is shown by the snapshots of the vertical component of relative vorticity at the ocean surface and the steric height in Figure 3. The relative vorticity field exhibits the attributes of eddies and filaments associated with meso- and sub-mesoscale turbulence in the model. The snapshot of steric height exhibits both large-scale features associated with mesoscale eddies, as well as radially-coherent features associated with propagating internal gravity waves and, specifically, the baroclinic tide.

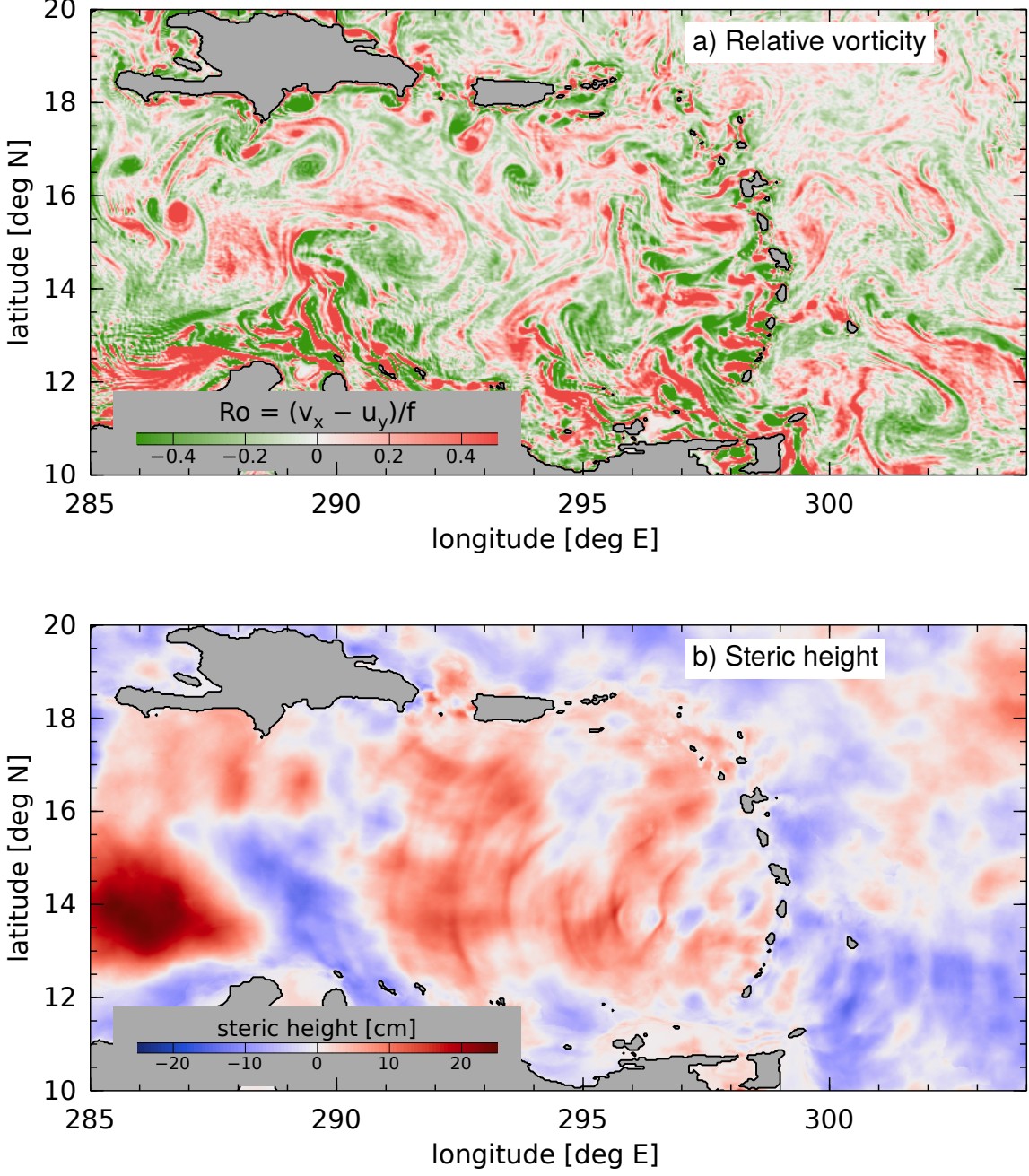

**Figure 3.** Snapshots of AMSEAS model outputs valid on 2013-01-02. (a) Relative vorticity field (expressed as a Rossby number) contains a spectrum of small circular eddies and filamentous structures. (b) Steric height anomaly (relative to the 2-yr. average).

# 3 Phase-locked tides in AMSEAS

The present effort analyzes AMSEAS output from the two-year period, January 2013 – December 2014. For a given date $T$, AMSEAS produces a nowcast valid at $T+0$ hr, and forecasts every 3 hours, up to $T+96$ hr, except for occasional interruptions. The nowcasts and forecasts consist of two-dimensional fields of sea level anomaly (SLA), $\eta_{ij}$, as well as three-dimensional fields of temperature, $T_{ijk}$, and salinity, $S_{ijk}$, on a fixed longitude-latitude-height grid, $(\lambda_i, \theta_j, z_k)$.

Identification of phase-locked tidal elevation requires considerable post-processing in order to separate the barotropic and baroclinic SLA components. The SLA output by AMSEAS is the sum of barotropic and baroclinic components, but these components may be separated, approximately, by computing a steric height anomaly from the vertical profiles of temperature and salinity which are also provided by AMSEAS. The steric height anomaly (hereafter referred to simply as the steric height) is defined as,

$$\eta'_{ij} = \sum_{k=1}^{N} \frac{\overline{\rho}_{ijk} - \rho(T_{ijk}, S_{ijk}, z_k)}{\rho_o} \Delta z_k, \tag{1}$$

where the density, $\rho(T, S, z)$, is computed using the equation of state of seawater (IOC et al., 2010), $\overline{\rho}_{ijk}$ is the time-average density, and $\rho_o = 1035$ kg/m$^3$ is a reference density. The quantity, $\eta'$, is the temporal anomaly of the steric height relative to the ocean bottom. Because of simplifications in the dynamics and thermodynamics of NCOM, as well as numerical truncation error (Mellor and Ezer, 1995; Greatbatch et al., 2001), the steric height computed in this manner does not agree precisely with other definitions of the baroclinic sea level, such as might be inferred from projecting the velocity field onto dynamical modes (Wunsch, 2013; Kelly, 2016).

The individual 96-hr forecasts produced by AMSEAS are too short for harmonic analysis to produce useful frequency resolution. Instead, the entire 2-year time series is subjected to harmonic analysis to estimate those tides which are phase-locked over the entire period. Because any given date-time may contain up to five different estimates for the steric height from the nowcast ($T + 0$ hr) and the forecasts (from $T + 3$ hr to $T + 96$ hr in three hour increments), the harmonic analysis is computed as an unweighted least-squares fit to all the data, including the overlapping forecasts. The following tides are included in the analysis: M$_2$, S$_2$, K$_2$, N$_2$, 2N$_2$, K$_1$, O$_1$, P$_1$, and Q$_1$ (provided through tidal open boundary conditions and the tide-generating force) and M$_3$, MS$_4$, M$_4$, and MN$_4$ over-tides (generated by the model's nonlinear dynamics). Note that S$_4$ and higher frequencies may be present in the model, but they are aliased by the 3-hour time sampling.

A detailed comparison between the modeled and observed phase-locked M$_2$ and K$_1$ tides at the sites indicated in Figure 2 is provided in Appendix A. The main features of the observed tide are reproduced in the model. These include a counter-clockwise propagation of the M$_2$ tide around an amphidrome in the northeast Caribbean and standing-wave-like behavior of K$_1$, exhibiting maximum amplitude of about 10 cm along the South American coast (Kjerfve, 1981; Torres and Tsimplis, 2011).

The curvature of the wave-like features in Figure 3b suggests that the baroclinic tide is generated at a relatively small number of sites, namely, at Mona Passage between the Dominican Republic and Puerto Rico (18°N, 292°E), and along the southern

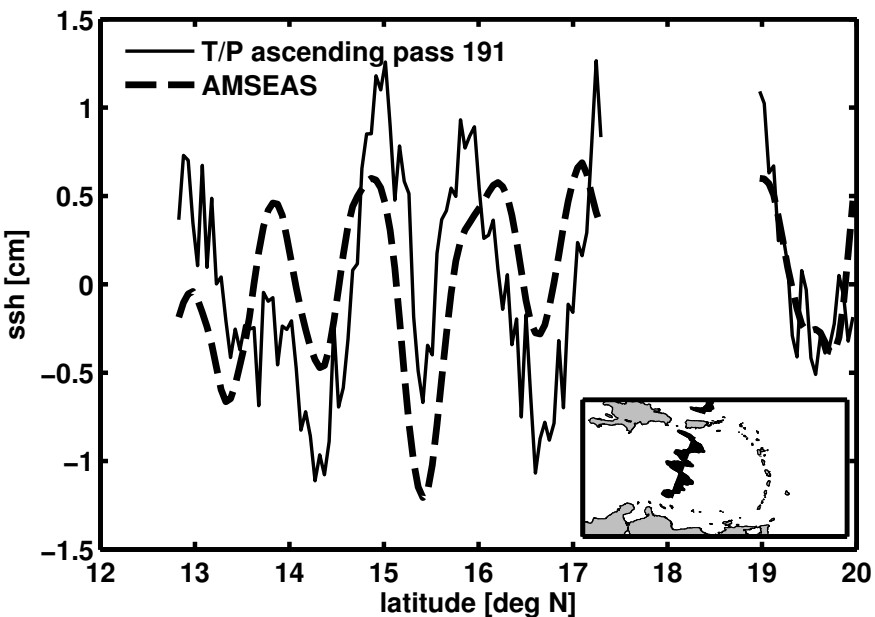

**Figure 4.** Internal tide along Jason-1 track #191. The in-phase component of the high-passed $M_2$ tide has maximum peak-to-peak amplitude of 1.5cm in AMSEAS (dashed), and slightly larger amplitude in the altimeter data (solid). Alignment of peaks and troughs is good, particularly near the apparent generation site, Mona Passage, west of Puerto Rico (inset).

Windward Islands (14°N, 298°E; see caption of Figure 2). Satellite altimeter ground tracks are not dense enough to map the baroclinic tides accurately in the Caribbean; however, the signals are unmistakable along individual tracks. Figure 4 compares the in-phase component of the $M_2$ tide in AMSEAS with the same quantity inferred from altimetry along a track which passes south of Mona Passage. The peaks and troughs are roughly aligned in the model and observations, and the amplitude of the
5   waves are quite similar in mid-basin, although the values certainly differ in detail.

The calibration of AMSEAS to improve its representation of baroclinic tides has not been attempted, and the effort involved in such a task would be substantial as it pertains to both the model itself and the choice of appropriate observational data. For example, it is likely that further refinement of the resolution of AMSEAS would improve the representation of the phase-locked tide. While the baroclinic tide SLA field is predominantly a mode-1 phenomena, with wavelength in excess of 100 km, achiev-
10   ing quantitative accuracy depends on resolving the detailed seafloor topography of the generation sites; based on experience modeling other sites, this requires a horizontal resolution of 1–2 km (Zaron and Egbert, 2006; Guihou et al., 2017; Aslam et al., 2018). Furthermore, the distinction between barotropic and baroclinic sea level is unambiguous in the model, but this same distinction cannot be made with most observations, such as altimetry, so calibration efforts would need to contend with the ambiguous attribution of model-data differences to barotropic, baroclinic, and non-tidal processes, as well as measurement
15   noise. Non-phase-locked tidal variability may also contribute to the differences between the model and observations, but to a degree which is presently unknown. Appendix A contains a graphical comparison of AMSEAS tides versus those inferred from altimetry.

The above remarks concerning the challenges of calibrating AMSEAS motivate the methodology used to assess the forecast error of non-phase-locked tides in AMSEAS. In the nomenclature of numerical weather forecasting, the approach taken is a self-analysis verification, or self-verification (e.g., Privé and Errico, 2015). The forecast error is measured by comparing the nowcast valid at date, $T_n$, with a forecast previously computed on the date, $T_f = T_n - \tau$, with a given lead-time, $\tau$, i.e.,

$T_f + \tau = T_n$. This approach may be contrasted with a forecast verification based on independent observations, in which the forecast and observations are compared directly as the latter become available. The self-verification is likely to lead to an optimistic estimate of the forecast error; nonetheless, it appears to be the only feasible approach to assessing the predictability of the non-phase-locked tide at present. The reasons for using the self-verification have more to do with the available data rather than the forecast system, since there are no data which can reliably distinguish non-phase-locked and phase-locked tidal

variability over the time and space scales represented within the AMSEAS forecasts.

## 4    Non-phase-locked tides

The previous section focused on the phase-locked tide, the unambiguous definition of which is provided by harmonic analysis and the constant phase with respect to the known astronomical tidal potential. In contrast, the definition of the non-phase-locked tide is potentially ambiguous since it involves distinguishing tide-band and non-tidal variability, which implicitly requires

either a dynamical definition or a definition based on frequency bandwidth. In addition, a bandwidth-based definition must be meaningful within the available duration of each forecast, which is only four days (from $T + 0$ to $T + 96$ hr). The challenge, then, is to develop a decomposition which can diagnose the non-phase-locked tide from a small number of snapshots of the steric height (e.g., Figure 3b).

An example of such a decomposition is presented in Figure 5, in which the steric height is represented as the sum of a

low-frequency component (Fig. 5a), a phase-locked tidal component (Fig. 5b), and a high-frequency component (Fig. 5c). The high-frequency component, which shall be identified with the non-phase-locked tide, is simply computed as the residual of total steric height, minus the predicted phase-locked tide, minus the low-frequency 24-hour-average steric height. More tersely, the non-phase-locked tide is defined as the anomaly of steric height with respect to the daily average of the de-tided steric height. To compute this quantity a tidal prediction is created using the complex harmonic constants for the 13 astronomically-forced

and compound tides described previously, and this predicted tide is removed from the steric height. Then, the daily average of the de-tided steric height is computed centered at noon of each forecast day ($T + 12$, $T + 36$, $T + 60$, and $T + 84$ hr) to provide an estimate of the low-frequency steric height field.

This methodology provides a pragmatic definition of the non-phase-locked tidal steric height valid at $T + 12$, $T + 36$, $T + 60$, and $T + 84$ hr. Alternative approaches could be envisioned, perhaps involving band-pass filtering or complex demodulation,

but they would be problematic due to phase errors near the start and end of the 4-day forecast windows. The present approach is relatively simple to implement and explain, and it unambiguously partitions the steric height variance between the low-frequency motion, phase-locked tides, and high-frequency processes, the latter being dominated by non-phase-locked tides (as may be verified a posteriori, e.g., Fig. 10). The fields centered at $T + 12$ hr shall be regarded as the verifying analyses (nowcast)

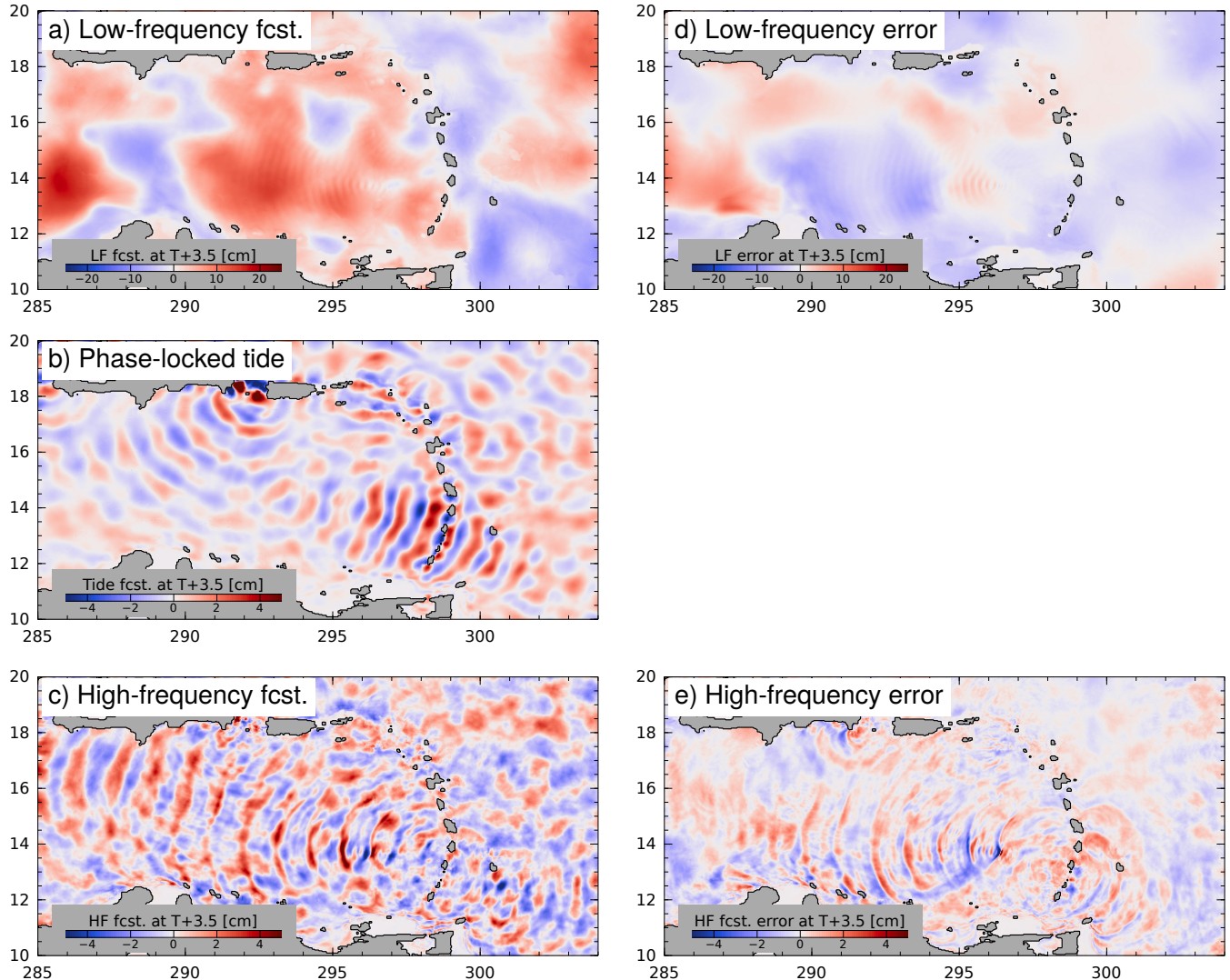

**Figure 5.** Forecast decomposition and errors for the date, 2013-01-04. For the purpose of analysis, the steric height forecast is decomposed into a sum of three components: (a) low-frequency (24-hour average), (b) phase-locked tide, and (c) high-frequency (residual). The forecast error is defined as the forecast minus the verifying analysis, valid at the same time. The two components of the forecast error are (d) the low-frequency component, and (e) the high-frequency component; the phase-locked tide is identical in the forecast and the analysis. Note different color scales shown for the high- and low-frequency components.

which are to be compared with the forecasts from 3 previous days, at $(T-24)+36$, $(T-48)+60$, and $(T-72)+84$. To shorten the notation, forecasts at these lead times will be denoted with the time offset in days, as "$T+1.5$", "$T+2.5$", and "$T+3.5$", in subsequent figures.

The decomposition of the steric height field just described is illustrated in Figure 5 for the representative date, 2013-01-04.
5 The low-frequency component of the forecast steric height (Fig. 5a) resembles a spatially-smoothed version of the snapshot

shown previously (Fig. 3b), but it is obtained by temporal, not spatial, averaging. The small-scale waves in Figure 3b are the sum of the predicted tide (Fig. 5b) plus the high-frequency residual (Fig. 5c). The panels in the left column of Figure 5 are valid on $T = 2013\text{-}01\text{-}04\text{T}12:00:00Z$ (i.e., at 1200 hours UT on 2013-01-04), but they were forecast 3.5 days prior, on $T_f = T - 84$ hr $= 2013\text{-}01\text{-}01\text{T}00:00:00Z$; the right panel shows the errors in the forecasts, computed by subtracting the

nowcast at $T_n = T$. The low-frequency error field (Fig. 5d) is relatively smooth and takes on largest values in the southeast, near the edge of the continental shelf where strong currents are present (13°N, 287°E; cf., Fig. 1). The error in the high-frequency component of the steric height (Fig. 5e) exhibits wavelike features. For this particular date, it is clear that the magnitudes of the forecast error fields (right panels) are smaller than the forecasts themselves (left panels); however, the error fields are non-random and display the features one might associate with difficult-to-forecast components of the flow field, e.g., small scales

and high-current zones. Note that the range of steric heights shown in panels (a) and (d) is different from that used in the other panels of Figure 5.

To illustrate the character of the forecast errors as a function of increasing lead time, $\tau$, Figure 6 shows the sum of the low- and high-frequency steric height errors for three lead times, $\tau = 1.5, 2.5,$ and 3.5 days, valid on the same date as above. The steric height signal associated with mesoscale features is in the range, $\pm 15$ cm, which is much larger than the typical range,

$\pm 5$ cm, associated with the baroclinic tide. The magnitude of the forecast error associated with the low-frequency flow is somewhat larger than the magnitude of the error associated with the high-frequency flow (cf., Fig. 5d and e), but their increase in time is evident in Figure 6.

Forecast errors of the low- and high-frequency steric height components exhibit different dynamical features. The most striking qualitative features of the high-frequency forecast and the high-frequency error (Fig. 5c and e) are spatially coherent

wave trains. Note that the peak amplitudes of both the low- and high-frequency forecasts are considerably larger than the error fields, thus, the model apparently captures much of the low-frequency variability that modulates the high-frequencies. The high-frequency error field is associated with small scales, and in Figure 5e it exhibits wavefronts which appear to radiate from Aves Escarpment (13.5°N,296°E) and the west side of Mona Passage (18°N,291°E). To gain some insight into the oceangraphic processess leading to these forecast errors, Figure 7 shows transects of high-frequency temperature (Fig. 7a-c)

and low-frequency Brunt-Vaisala frequency anomaly (Fig. 7d). The high-frequency steric height along this section exhibits a wave with nearly 10 cm amplitude at 296°E, in both the forecast and the verifying analysis (Fig. 7a,b). The error field (Fig. 7c) shows that this feature is slightly offset in the analysis and forecast; however, the sawtooth shape of the steric height waveform leads to an error field with nearly the same peak amplitide as in the forecast. This appears to be a mode-1 internal wave which has steepened considerably, possibly during its passage over Aves Ridge. The error in the low-frequency forecast (Fig. 7d),

expressed in terms of the error in the buoyancy frequency, $N(z)$, is not pronounced over the Ridge, rather, it displays a broad positive anomaly at about 150 m depth and a negative anomaly at the base of the surface mixed layer.

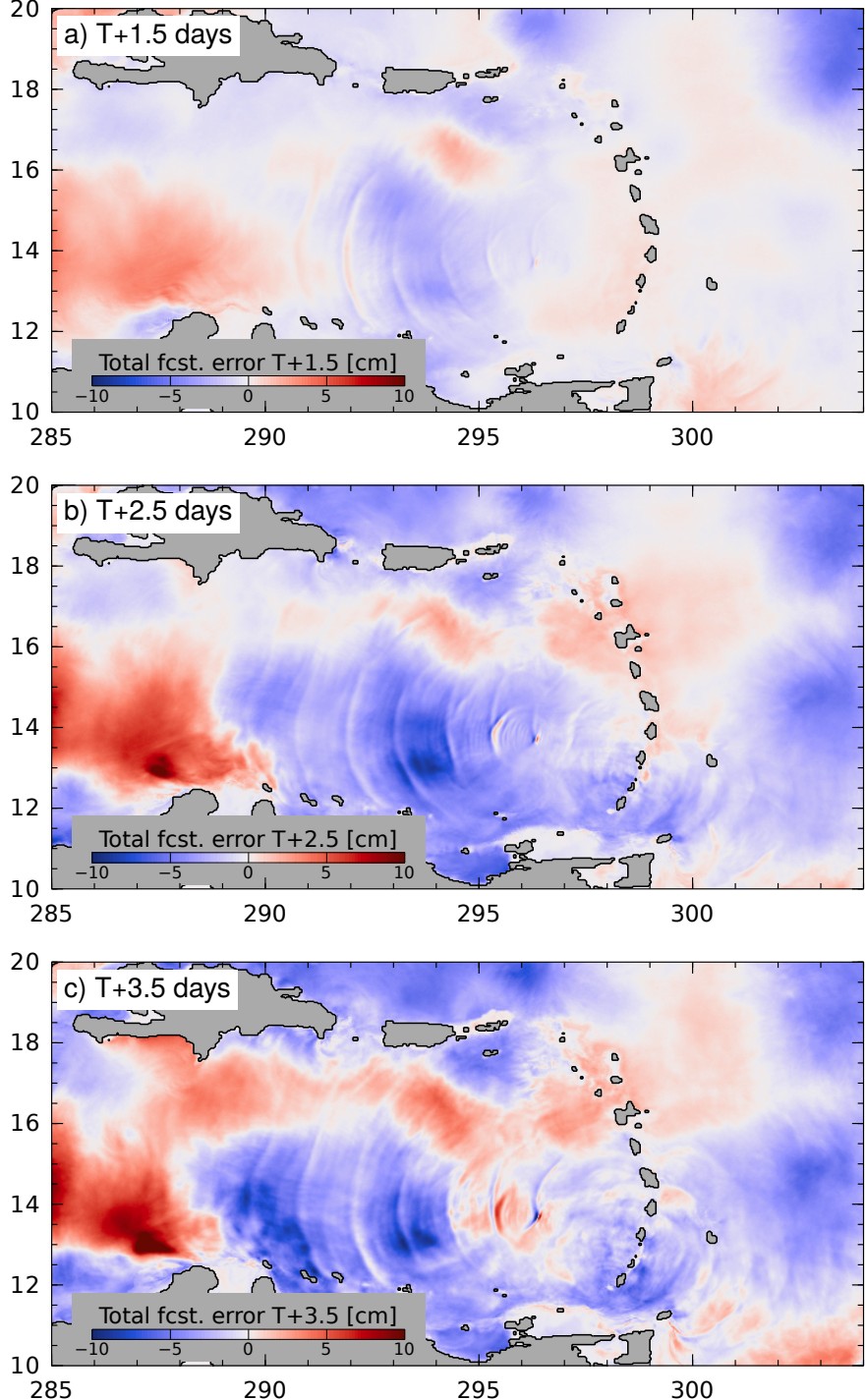

**Figure 6.** The steric height forecast error for different lead times, for $T_n = 2013\text{-}01\text{-}04T12:00:00Z$: (a) $T_n = T_f + 1.5$ days, (b) $T_n = T_f + 2.5$ days, and (c) $T_n = T_f + 3.5$ days. Note that the color scale differs from previous figures; it was chosen to make the error growth visible as a function of increasing forecast lead time.

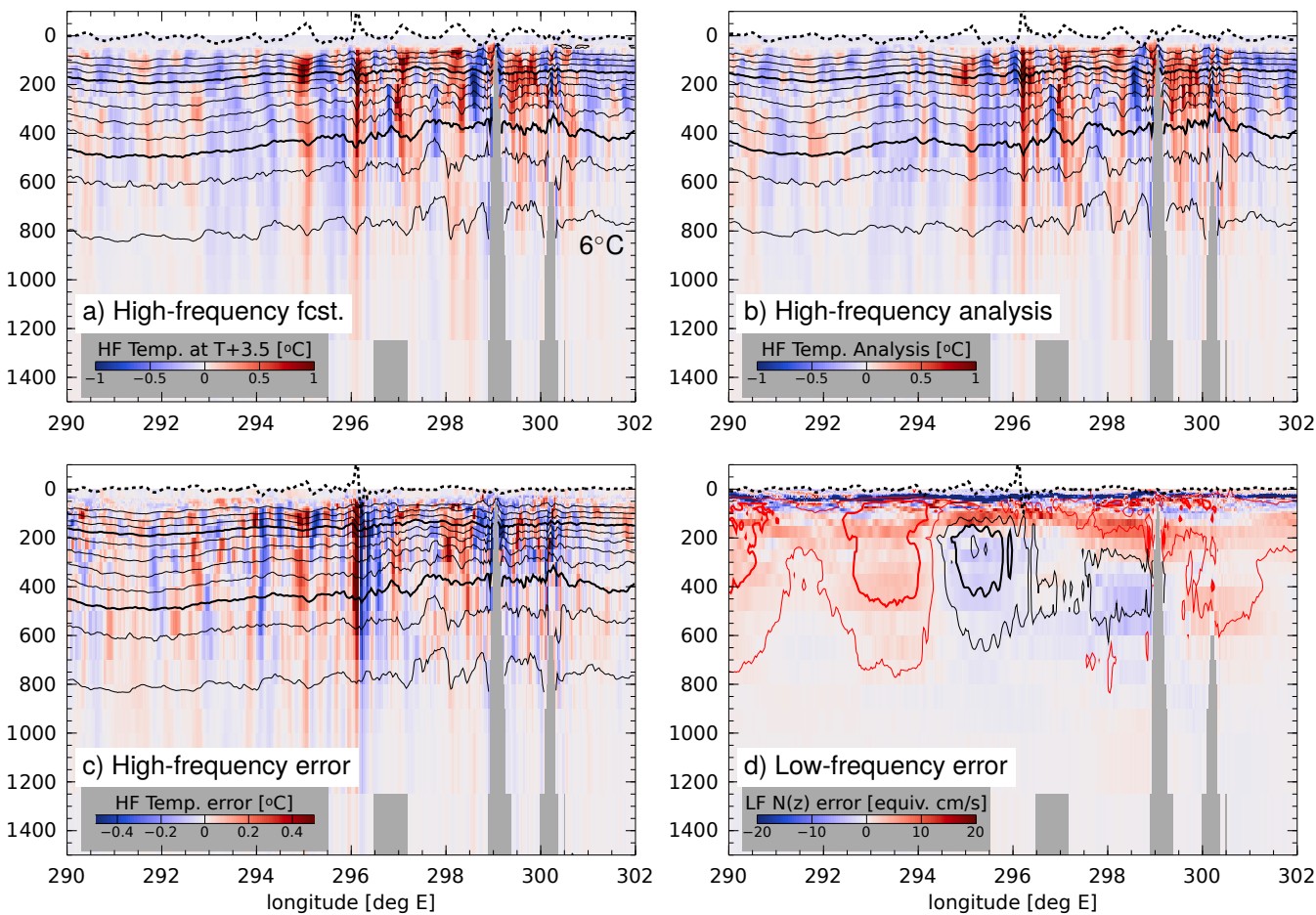

**Figure 7.** Zonal transect of temperature at 13.7°N (c.f., Figure 5). (a) The high-frequency (HF) temperature forecast, and (b) the HF verifying analysis, both valid at $T + 3.5 = 2013\text{-}01\text{-}04T12:00:00Z$. The solid contours indicate the total temperature field (2°C contour interval). The dashed line at the top, centered on $z = 0$, is the HF steric height in units of millimeters (the maximum at $x = 296°E$ is almost $\eta' = 10\text{cm}$). This transect cuts slightly north of the shallow seamount, Aves Ridge, which rises to within about 600m of the ocean surface at 13.4°N, 297°E. Here the topography is represented by the gray shading visible near 297°E, 299°E, and 300°E. (c) The HF forecast temperature error (colorscale) and steric height error (dashed line centered at $z = 0$) is largely associated with a slight shift in the location of the wave at 296.1°E. (d) Error in the LF forecast is represented with error in buoyancy frequency, $N(z)$, which is multiplied here by 200m to scale it like mode-1 baroclinic wave phase speed (based on the WKB-approximation, $c_1 = \pi^{-1} \int_{-H}^{0} N(z)dz$, about 3m/s). Contours represent LF temperature error (0.5°C contour interval, positive in black, negative in red) with thicker contours at $+0.75$°C and $-0.75$°C.

## 5   Results

Given the above decomposition of the nowcast/forecast steric height fields, the statistics of the forecast errors have been computed for the two year period, 2013-2014.

A summary of the spatial statistics is provided in Figure 8, which shows the root-mean-square of the different steric height components, corresponding to the panels in Figure 5. The low-frequency forecast (Fig. 8a) and its error (Fig. 8d) are spatially uniform except for the influence of water depth on steric height variability. The spatial distributions of the phase-locked

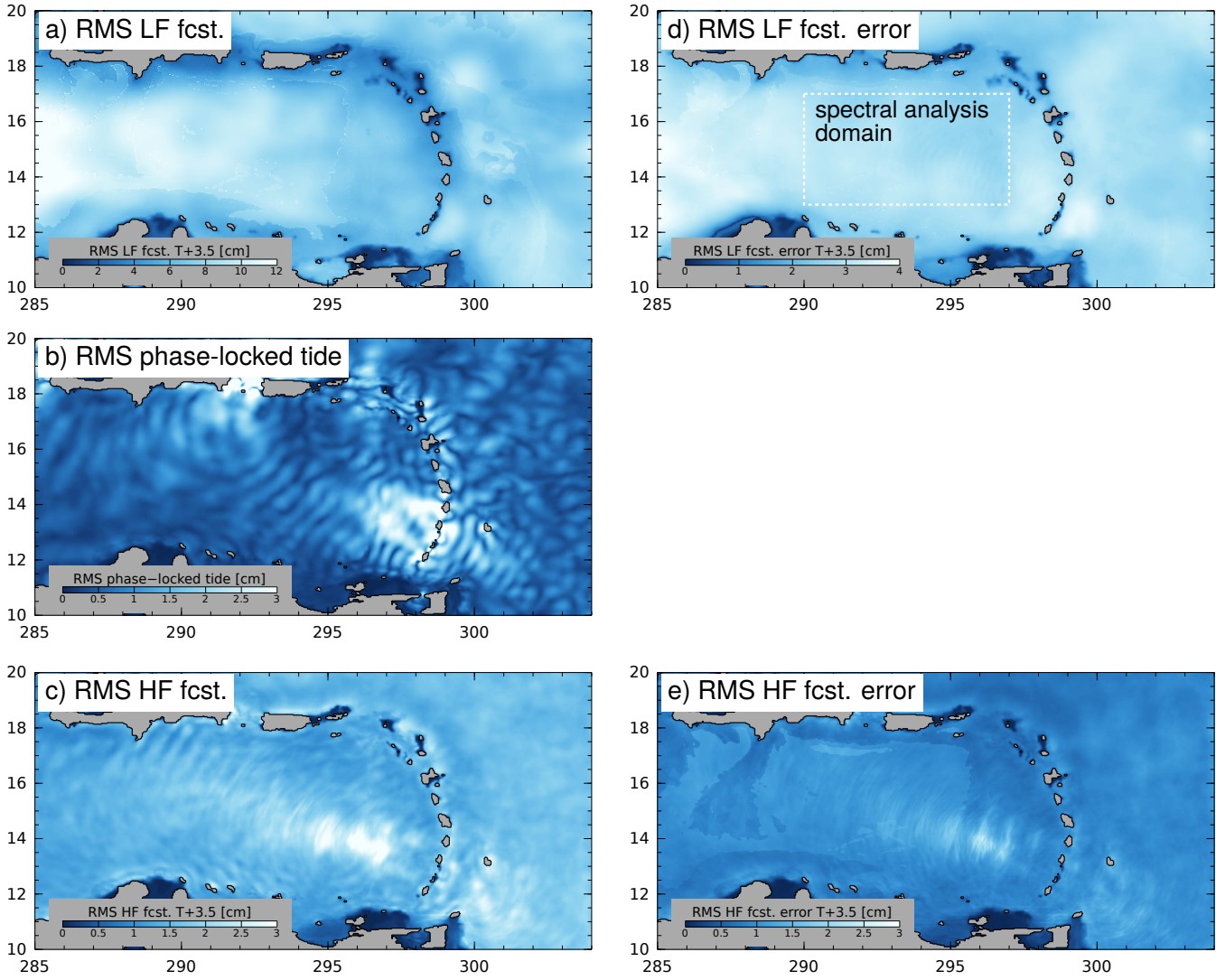

**Figure 8.** Root-mean-square (RMS) quantities valid at $T_f + 3.5$ days (panel layout corresponds to snapshots in Figure 5): (a) low-frequency forecast, (b) phase-locked tide forecast, and (c) high-frequency forecast; and the corresponding errors: (d) low-frequency forecast error, and (e) high-frequency forecast error. Note the different colorscales used for low-frequency variability. The "spectral analysis domain" in (d) indicates the region used for the computation of the radial wavenumber spectra in Figure 10, below; it is also the region of spatial averaging for the forecast errors summarized in Figure 9.

(Fig. 8b) and non-phase-locked tides differs (Fig. 8c); the amplitude of the phase-locked tide is largest near its sources in Mona Passage and near the Windward Islands, while the non-phase-locked tide is largest around Aves Escarpment, some distance from the wave source. The high-frequency forecast error (Fig. 8e) is spatially uniform, except near Aves Ridge. This pattern might be explained by statistically homogeneous refraction of the internal tide by the mesoscales, except in the vicinity of

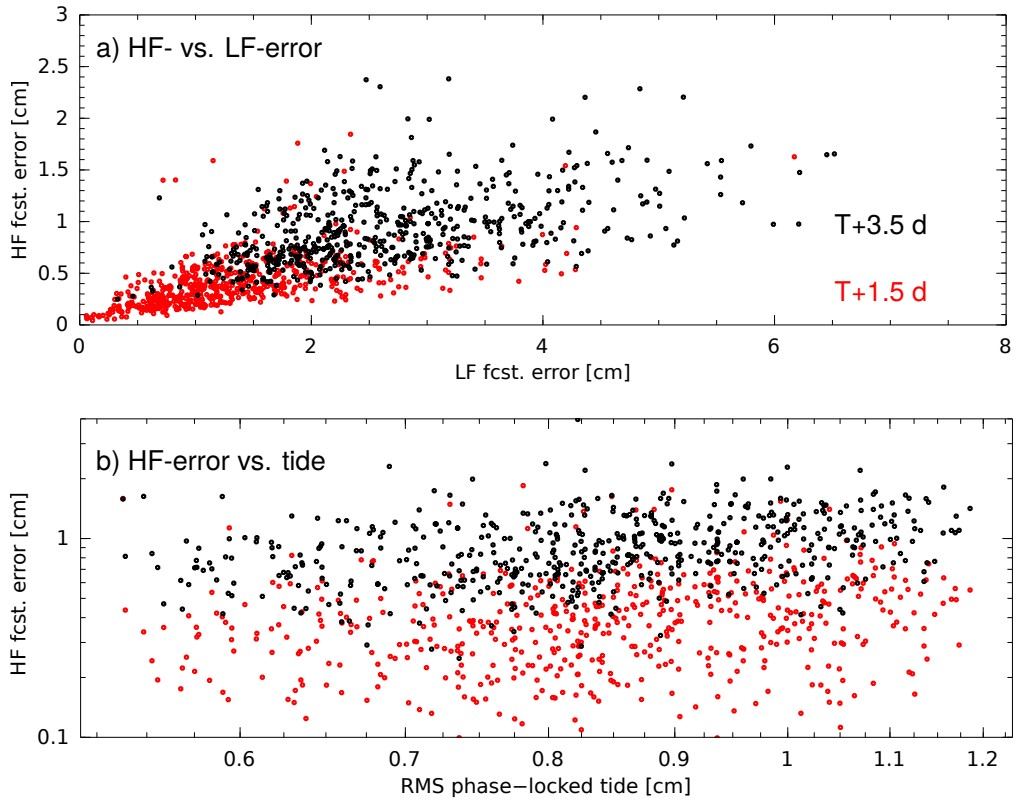

**Figure 9.** High-frequency (HF) forecast error. (a) HF forecast error is an increasing function of LF forecast error, and error increases as a function of lead time (T+1.5 days shown in red, T+3.5 days shown in black). (b) HF forecast error depends weakly on the phase-locked tide (red and black dots as in (a)).

Aves Ridge where the internal tide generated near the Windward Islands is focused (Fig. 8b,c), and where sawtooth-shaped (nonlinear) wave profiles were noted above (c.f., Fig. 7).

Because the non-phase locked tide arises as a consequence of the time-variable propagation medium, it was hypothesized that the forecast error for the non-phase-locked tide would be related to the forecast error for the low-frequency flow. This hypothesis is generally confirmed by the statistics in Figure 9a, which shows the high-frequency error as a function of the low-frequency error. The forecast errors for the two components of the steric height are positively correlated (note that the errors reported here are root-mean-square averages over the domain denoted, "spectral analysis domain" in Figure 8d); however, there is considerable scatter in the high-frequency error which is unrelated to the low-frequency error. A plot of the forecast error versus the amplitude of the phase-locked tide (Fig. 9b) indicates that the error is only weakly dependent on the phase-locked tide (e.g., the spring-neap cycle). The scatter of the high-frequency error with respect to these root-mean-square statistics is consistent with the genesis of the errors at small-scale features which are inherently less predictable and constrained by observations. Note that the errors for the $T + 3.5$ day forecasts (black dots) are larger than for the $T + 1.5$ day forecasts (red dots), as would be expected (cf., Fig. 6).

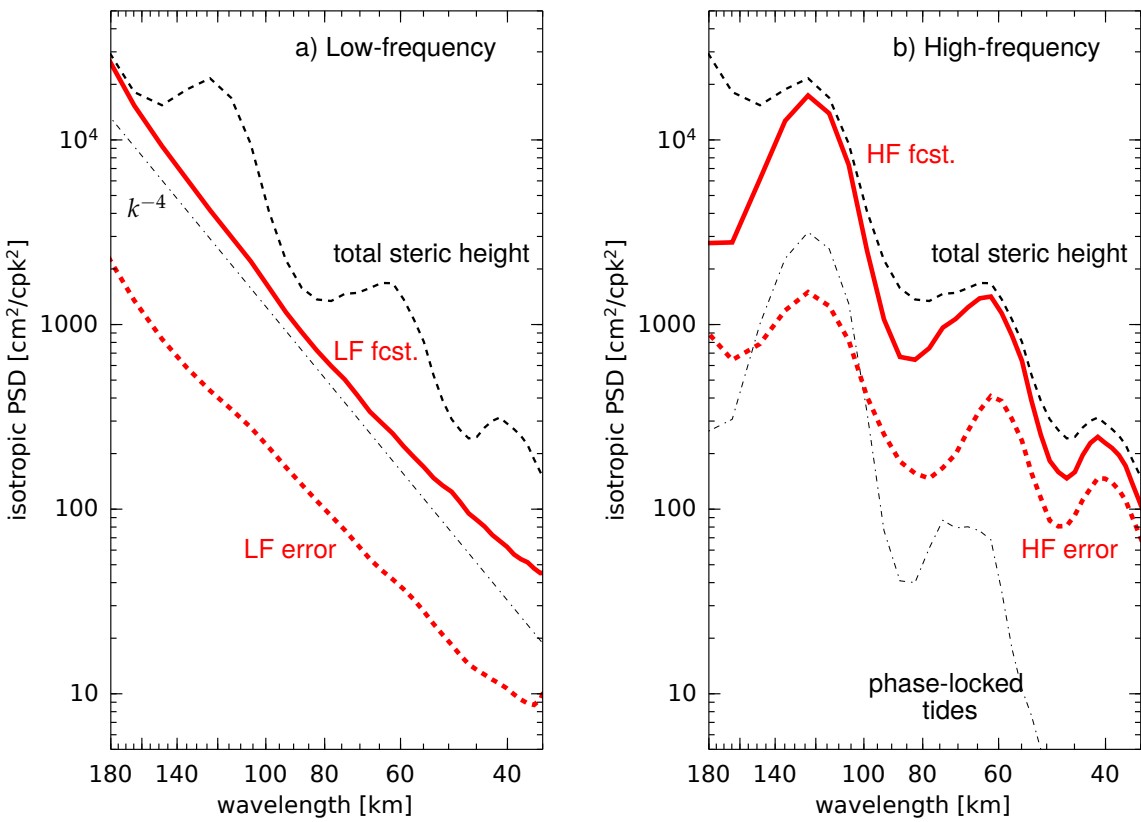

**Figure 10.** Isotropic wavenumber power spectral density for (a) low-frequency, and (b) high-frequency steric height components. The $k^{-4}$ wavenumber dependence (dash-dot black line) is shown for reference in panel (a); the spectrum of the phase-locked tides (dash-dot black line, a component of the HF fcst.) is shown for reference in panel (b); and the total steric height spectrum is repeated in both panels (dashed black line). The LF error (dashed red line, panel (a)) is approximately a constant fraction of the LF forecast (solid red line) independent of scale. In contrast, the HF error (dashed red line, panel (b)) is an increasing fraction of the HF forecast (solid red line) as wavelength decreases.

A spectral analysis of the steric height anomaly has been conducted in order to understand the scale-dependence of the forecast error. The isotropic power spectral density shown in Figure 10, equal to $(2\pi k)^{-1}$ times the azimuthally averaged 2-dimensional power spectrum (within the "spectral analysis domain" indicated in Fig. 8d), clearly exhibits peaks related to the baroclinic tide. The tidal peaks are completely absent in the low-frequency forecast and its error (solid and dashed red,

5  respectively, in Fig. 10a). In contrast, the tidal peaks dominate the spectra of the high-frequency steric height (Fig. 10b). The variance associated with the mode-1 semidiurnal baroclinic tide, at a wavelength of 120 km, is partly associated with the predictable phase-locked tide (dash-dot black line), and the forecast error (dashed red line) is less than 10% of the total high-frequency forecast variance (solid red line). At wavelengths shorter than the mode-1 wave, the high-frequency forecast error is larger than the phase-locked tide and it is a larger fraction of the total high-frequency variance. In other words, as would be

10  expected, the baroclinic tides are increasingly less phase-locked, and less predictable, at smaller scales.

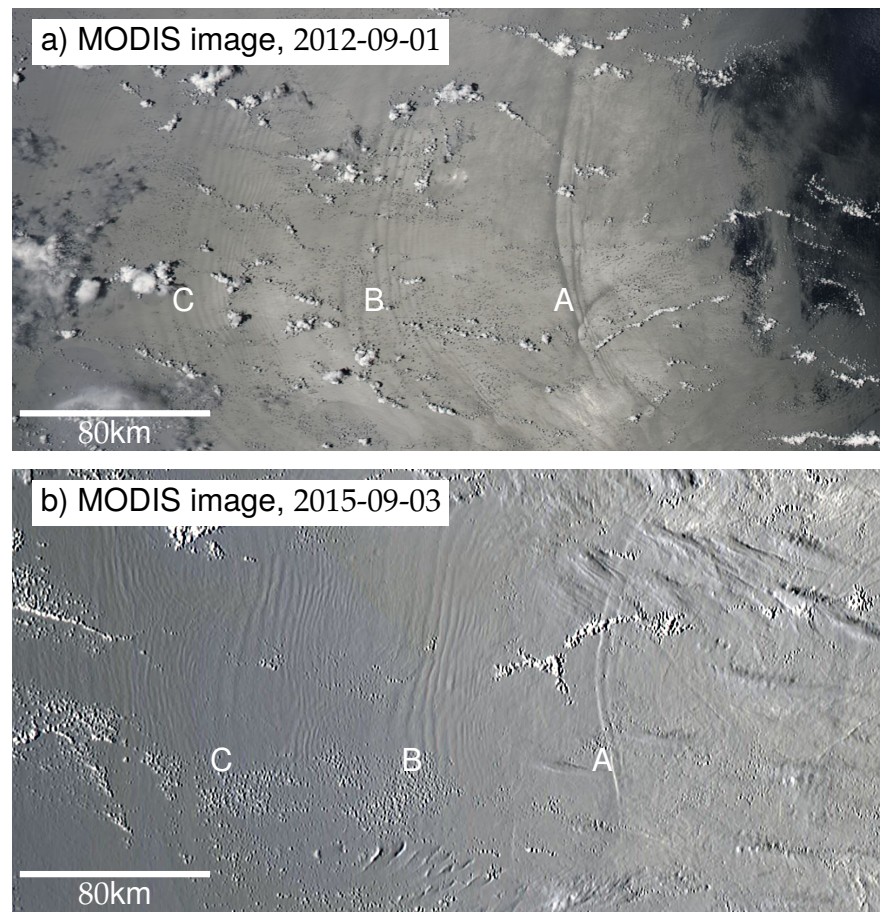

**Figure 11.** Internal wave packets in MODIS imagery from Aves Escarpment. MODIS sun-glint imagery from dates in (a) 2012 and (b) 2015 (digitally enhanced) exhibits the surface expression of three packets of internal waves, at longitudes marked A, B, and C. Remarkably, similar wave packets are visible in imagery from 2018-09-05 (not shown). The largest packet, A, is located at Aves Ridge ($14°20'$N, $296°30'$E) on Aves Escarpment.

## 6  Discussion

The results of this study provide a descriptive snapshot of AMSEAS forecast skill during one 2-year period, and they are presumably dependent upon the particular ocean observing system, numerical model resolution and configuration, and data assimilation algorithm used during this period. Nonetheless, the results provide some insight into the capabilities and limitations of such a system for predicting and explaining high-frequency ocean variability. For example, it is evident in Figure 10b that the wavelength peak associated with the mode-2 phase-locked tides (dashed dot black line) occurs at a slightly longer wavelength than the nearby peaks in either the high-frequency forecast (red line) or the forecast error (dashed red line). In fact, an examination of the model output (not shown) indicates that the high-frequency forecast and forecast error associated with the 60 km wavelength peak is related to the non-linear mode-1 baroclinic tide generated on the shoals of the Aves Escarpment,

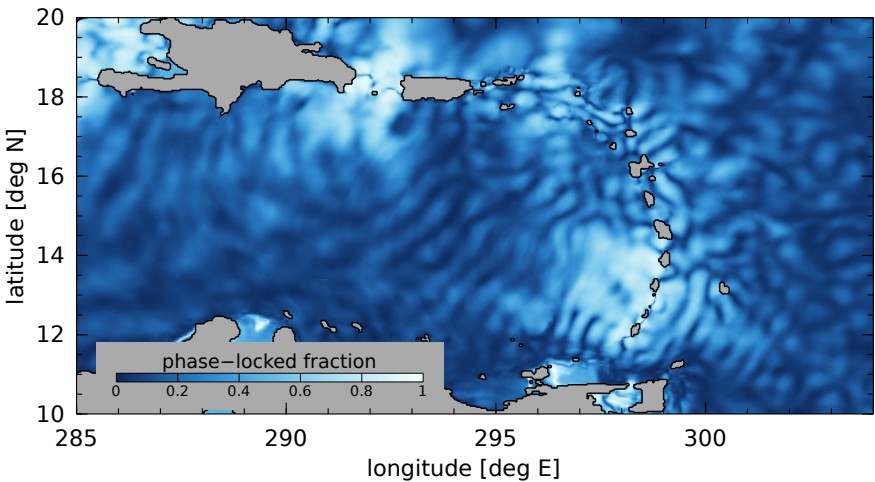

**Figure 12.** Fraction of variance, phase-locked versus total high-frequency variance (phase-locked plus non-phase-locked variance).

while the 70 km peak in the phase-locked tide is related to the mode-2 baroclinic tide. The nonlinear mode-1 dynamics may be AMSEAS' rendition of the internal wave packets generated along Aves Escarpment which have been identified in satellite sun-glint imagery (Alfonso-Sosa, 2013) and are the cause of coastal seiches around Puerto Rico (Giese et al., 1990; Alfonso-Sosa, 2015; Woodworth, 2017), nearly 1000 km to the northwest. Figure 10b indicates that a considerable fraction of the variance

associated with the wavenumber peak near 60 km is predictable by the AMSEAS system, even though is it not phase-locked with the astronomical tidal forcing. Indeed, a brief search of the MODIS sun-glint imagery found the pair of images displayed in Figure 11 in which similar wavepackets are found in images taken 3 years apart. These images provide evidence of nonlinear internal waves too small to be directly resolved by AMSEAS; however, their recurring pattern is certainly suggestive of some degree of predictability.

A dedicated study of the dynamics responsible for both the predictable and unpredictable high-frequency variability would be needed to understand the factors responsible for the time-variable propagation of the baroclinic tide, and its generation. The high-frequency forecast error is correlated with the low-frequency forecast error (Fig. 9), but the reason for this correlation has not been examined in detail. It was initially hypothesized that errors in location or intensity of (low-frequency) forecast mesoscale eddies would be the cause of errors in the high-frequency forecasts; however, there are other possible explanations

for this relationship. For example, the generation of higher baroclinic modes and nonlinear overtides near Aves Ridge probably depends on the near-surface stratification at this site, and there are many factors which might influence this, aside from the location of mesoscale eddies. Another explanation for the correlated errors is related to how the observed SLA data are assimilated with the NCODA system, which projects the observations onto sub-surface profiles of temperature and salinity. The presence of non-phase-locked tidal signals in the observations might be erroneously attributed to mesoscale features and

assimilated into the model; in this case, the low-frequency error would be caused by the high-frequency variability, rather than vice versa.

Efforts to map the baroclinic tide with satellite altimeter data are generally only capable of identifying the phase-locked part of the tidal signals (Ray and Mitchum, 1996; Carrère et al., 2004). Attempts to quantify the energy transport, and dissipation, of the baroclinic tide from altimetry data must make assumptions about the partitioning of energy between the phase-locked and non-phase-locked tides; however, present estimates for non-phase-locked tidal steric height suffer from sampling limitations (Zilberman et al., 2011; Kelly et al., 2015; Zaron, 2015, 2017). Ocean models are increasingly being used to study this quantity (Shriver et al., 2014; Kerry et al., 2016; Savage et al., 2017; Ansong et al., 2017), and AMSEAS is presently one of the highest-resolution models which includes both tides and wind-driven circulation. The ratio of phase-locked to total high-frequency steric height variance shown in Figure 12 indicates that the phase-locked tide dominates the variance – hence energy – only within a few hundred kilometers of the generation sites in the Caribbean Sea. The baroclinic tide in the middle of the Caribbean Sea is dominated by the non-phase-locked component. Whether or not this result is more broadly applicable is unknown, but the variance fraction is a useful diagnostic of the partition between open-ocean versus boundary dissipation of the baroclinic tide (Zaron, 2019).

Finally, an important caveat with the present study is related to the use of self-verifying analyses to assess the forecast errors. This approach is capable of measuring the predictability of the high-frequency steric height only to the extent that AMSEAS accurately represents the ocean dynamics. This approach is not capable of identifying systematic errors shared by the analysis and the forecast (e.g., errors in the phase-locked tides) and it thus provides a lower bound on the actual forecast error. To estimate this quantity, consider the root-mean-square goodness-of-fit to the assimilated SLA, which is roughly $\sigma = 6$ cm (Zaron et al., 2015). Assuming this value is the sum of independent components of instrumental measurement error, $\sigma_e = 3$ cm; baroclinic tide signal, $\sigma_t = 4$ cm (treated as "representation error" in the NCODA assimilation); and unknown analysis error, $\sigma_a$; one may use $\sigma_e^2 + \sigma_t^2 + \sigma_a^2 = \sigma^2$ to estimate $\sigma_a = 3$ cm. This is a lower bound for the analysis error at the measurement sites, as it does not account for the larger error between the satellite ground tracks. If $\sigma_a = 3$ cm is taken as the lower bound on the low-frequency forecast error, then Figure 9 suggests that $0.75$ cm is a lower bound on the high-frequency forecast error. This is a crude estimate which does not account for the geographic variability of either the ocean dynamics or the ocean observing system, but it provides a guide to the best case which may be attained in practice.

# 7 Conclusions

This study has examined the predictability of non-phase-locked baroclinic tides using four-day ocean forecast products from the AMSEAS system. It was motivated by the desire to understand our present capability for predicting baroclinic tidal SLA, which is expected to be a key limitation for measuring mesoscale and submesoscale processes with the forthcoming SWOT wide-swath satellite altimeter (Callies and Wu, 2019). The AMSEAS system is well-suited to this task since it represents the state-of-the-art among data-assimilating ocean forecast systems; it has resolution sufficient to realistically represent baroclinic tide generation and propagation, and the four-day forecast cycle is adequate to separate the SLA signals of low-frequency balanced motions from high-frequency processes, the latter being dominated by baroclinic tides.

The findings of this study indicate that a substantial fraction of non-phase-locked tidal sea level variability may be predictable with an ocean forecasting system. This result has been demonstrated using self-verifying analyses with the AMSEAS system, and future studies should investigate how to use independent high-frequency data to verify forecast skill statistics. The small scale and spatial inohomogeneity of the baroclinic tide makes this a challenging task. Nonetheless, many applications of

altimetry which regard the tidal SLA as noise would benefit from improved predictions of baroclinic tidal sea level.

*Code and data availability.*    The software and data products used in this research are available from the author.

## Appendix A:  Comparisons of observed and predicted phase-locked tides in the Caribbean

### A1    Tide gauges

Tables A1 and A2 provide quantitative comparisons of the phase-locked tide sea surface height at tide gauges (and one bottom

pressure sensor, DART-42407) in the Caribbean Sea for the main semi-diurnal and diurnal tides, $M_2$ and $K_1$. The observed data come from three sources, the National Data Buoy Center (for the DART buoy), the NOAA Center for Operational Oceanographic Products, and historical stations in Kjerfve (1981), as indicated in Table A1.

The amplitude and Greenwich phase lag of the observed (obs.) and predicted (pred.) AMSEAS sea surface heights are listed. The station number in the first column corresponds to the location in Figure 2. Stations located too close to distinguish on the

map are indicated with lower-case letters in the Tables, e.g., "7a Barbuda" and "7b Antigua".

Over-all, the observed and predicted tides agree within a centimeter or two at most stations, with a few interesting exceptions. Because tides are relatively small in the Caribbean, the $M_2$ fractional errors are large, particularly in the northeast (e.g., stations 3, 5, and 6), near the $M_2$ amphidromic point (Kjerfve, 1981). Because the tide gauge measurements cannot distinguish barotropic and baroclinic sea level, it may be that much of the error can be attributed to small scale differences in the baroclinic

tide. The amplitude errors for $K_1$ are small, less than a centimeter almost everywhere, but a widespread phase error is present, except, oddly, at stations 8a and 8b on Guadaloupe. The reason for the $K_1$ phase error is obscure, but it may be related to the fact that $K_1$ behaves like a standing wave, so its phase may be strongly influenced by wave reflection at both open boundaries and the coastline in the AMSEAS model.

**Table A1.** Tide gauge stations and tidal statistics: $M_2$

| | station | location [°W, °N] | H [cm]/ G [°] obs. | pred. | $\Delta$, pred.-obs. |
|---|---|---|---|---|---|
| 1 | DART-42407[a] | (68.22,15.29) | 2.5/134 | 3.1/145 | 0.6/10 |
| 2 | Mona Island, Puerto Rico[b] | (67.94,18.09) | 1.5/011 | 0.3/053 | -1.2/42 |
| 3a | Magueyes Island, Puerto Rico[b] | (67.05,17.97) | 0.8/023 | 1.5/106 | 0.7/83 |
| 3b | Penuelas, Puerto Rico[b] | (66.76,17.97) | 0.8/039 | 0.1/316 | -0.7/-83 |
| 4 | San Juan, Puerto Rico[b] | (66.12,18.46) | 15.9/020 | 15.5/016 | -0.4/-4 |
| 5a | St. Thomas Harbor, USVI[c] | (64.93,18.33) | 3.6/027 | 8.0/012 | 4.4/-15 |
| 5b | Cruz Bay, USVI[c] | (64.80,18.34) | 0.4/284 | 7.8/012 | 7.4/88 |
| 5c | Lameshur Bay, USVI[b] | (64.72,18.32) | 4.7/024 | 3.6/021 | -1.1/-3 |
| 6a | Lime Tree Bay, USVI[b] | (64.75,17.69) | 1.3/328 | 1.2/089 | -0.1/121 |
| 6b | Christiansted Harbor, USVI[b] | (64.70,17.75) | 3.5/346 | 2.9/355 | -0.6/9 |
| 7a | Barbuda[b] | (61.82,17.59) | 3.8/287 | 3.3/308 | -0.5/21 |
| 7b | Antigua[c] | (61.79,17.16) | 4.8/256 | 3.2/266 | -1.6/10 |
| 8a | Sainte Rose, Guadaloupe[c] | (61.70,16.33) | 6.5/241 | 5.0/229 | -1.5/-12 |
| 8b | Pointe-a-Pitre, Guadaloupe[c] | (61.53,16.23) | 8.7/233 | 5.8/227 | -2.9/-6 |
| 9 | Dominica[c] | (61.40,15.29) | 11.7/216 | 7.5/214 | -4.2/-2 |
| 10 | Forte de France, Martinique[c] | (61.05,14.58) | 5.5/211 | 9.7/212 | 4.2/1 |
| 11 | Castries, St. Lucia[c] | (61.00,14.02) | 9.4/191 | 10.7/209 | 1.3/18 |
| 12 | Barbados[c] | (59.63,13.10) | 22.9/227 | 19.5/216 | -3.4/-11 |
| 13 | Tobago Cays, Grenadines[c] | (61.35,12.63) | 12.2/210 | 13.2/208 | 1.0/-2 |
| 14 | Tobago[c] | (60.73,11.18) | 29.3/218 | 23.6/212 | -5.7/-6 |
| 15 | Nariva River, Trinidad[c] | (61.03,10.40) | 34.7/220 | 27.4/223 | -7.3/3 |
| 16 | Gaspar Grande, Trinidad[c] | (61.65,10.67) | 29.3/225 | 15.9/250 | -13.4/25 |
| 17 | Carupano, Venezuela[c] | (63.25,10.67) | 11.3/190 | 7.5/209 | -3.8/19 |
| 18a | Amuay, Venezuela[c] | (70.22,11.75) | 10.3/252 | 3.9/238 | -6.4/-14 |
| 18b | Zaparita, Venezuela[c] | (71.65,11.02) | 42.2/270 | 14.5/288 | -27.7/18 |
| 19 | Oranjestad, Aruba[c] | (70.05,12.52) | 4.0/161 | 4.5/158 | 0.5/-3 |
| 20 | Rio Hacha, Columbia[c] | (72.92,11.55) | 6.6/127 | 7.5/140 | 0.9/13 |

[a] National Data Buoy Center, station 42407, http://www.ndbc.noaa.gov.
[b] NOAA Center for Operational Oceanographic Products and Services, http://tidesandcurrents.noaa.gov.
[c] Kjerfve (1981).

**Table A2.** Tide gauge stations and tidal statistics: $K_1$

| | station | location [°W, °N] | H [cm]/ G [°] | | |
|---|---|---|---|---|---|
| | | | obs. | pred. | Δ, pred.-obs. |
| 1 | DART-42407 | (68.22,15.29) | 8.8/236 | 9.0/221 | 0.1/-15 |
| 2 | Mona Island, Puerto Rico | (67.94,18.09) | 9.3/237 | 8.8/218 | -0.5/-19 |
| 3a | Magueyes Island, Puerto Rico | (67.05,17.97) | 8.0/233 | 8.3/218 | 0.3/-15 |
| 3b | Penuelas, Puerto Rico | (66.76,17.97) | 8.2/234 | 8.2/221 | -0.0/-13 |
| 4 | San Juan, Puerto Rico | (66.12,18.46) | 9.0/228 | 8.9/215 | -0.1/-13 |
| 5a | St. Thomas Harbor, USVI | (64.93,18.33) | 8.0/233 | 8.1/210 | 0.1/-23 |
| 5b | Cruz Bay, USVI | (64.80,18.34) | 7.2/220 | 8.2/211 | 1.0/-9 |
| 5c | Lameshur Bay, USVI | (64.72,18.32) | 8.3/229 | 8.4/215 | 0.1/-14 |
| 6a | Lime Tree Bay, USVI | (64.75,17.69) | 8.3/237 | 8.4/220 | 0.1/-17 |
| 6b | Christiansted Harbor, USVI | (64.70,17.75) | 8.2/229 | 8.4/215 | 0.2/-13 |
| 7a | Barbuda | (61.82,17.59) | 7.2/229 | 7.7/216 | 0.5/-13 |
| 7b | Antigua | (61.79,17.16) | 7.6/234 | 7.9/217 | 0.3/-17 |
| 8a | Sainte Rose, Guadaloupe | (61.70,16.33) | 7.7/218 | 8.2/219 | 0.5/1 |
| 8b | Pointe-a-Pitre, Guadaloupe | (61.53,16.23) | 7.3/220 | 8.2/219 | 0.9/-1 |
| 9 | Dominica | (61.40,15.29) | 10.2/231 | 8.5/221 | -1.7/-10 |
| 10 | Forte de France, Martinique | (61.05,14.58) | 7.8/246 | 8.7/221 | 0.9/-25 |
| 11 | Castries, St. Lucia | (61.00,14.02) | 8.6/238 | 8.9/222 | 0.3/-16 |
| 12 | Barbados | (59.63,13.10) | 9.3/239 | 8.9/222 | -0.4/-17 |
| 13 | Tobago Cays, Grenadines | (61.35,12.63) | 10.4/243 | 9.2/224 | -1.2/-19 |
| 14 | Tobago | (60.73,11.18) | 9.6/244 | 9.6/225 | -0.0/-19 |
| 15 | Nariva River, Trinidad | (61.03,10.40) | 7.6/247 | 10.3/229 | 2.7/-18 |
| 16 | Gaspar Grande, Trinidad | (61.65,10.67) | 6.7/244 | 11.2/233 | 4.5/-11 |
| 17 | Carupano, Venezuela | (63.25,10.67) | 10.4/238 | 11.0/231 | 0.6/-7 |
| 18a | Amuay, Venezuela | (70.22,11.75) | 12.4/243 | 11.6/232 | -0.8/-11 |
| 18b | Zaparita, Venezuela | (71.65,11.02) | 13.6/250 | 13.7/241 | 0.1/-9 |
| 19 | Oranjestad, Aruba | (70.05,12.52) | 9.0/241 | 9.6/224 | 0.6/-17 |
| 20 | Rio Hacha, Columbia | (72.92,11.55) | 9.1/240 | 9.3/226 | 0.2/-14 |

## A2 Satellite altimetry

A graphical comparison of AMSEAS and altimeter-derived tides is provided in Figures A1 and Figures A2. The $M_2$ tide was estimated from Topex/Poseidon (thin black line) and Jason-1 (thin blue line) altimetry data for the periods, 1992–2002 and 2002–2009, respectively, while the AMSEAS phase-locked tide was computed as described in Section 3 of the main text.

There are two primary inferences to be drawn from the comparisons. The first pertains to the data; namely, there are differences in the harmonic constants estimated from the Topex/Poseidon and Jason-1 data. It is not clear at present whether these differences are caused by long-term trends and interannual variability of the tides (e.g., Müller, 2011; Devlin et al., 2017), or by noise related to mesoscale variability (e.g., Ray and Byrne, 2010). Differences in the tide are at the level of roughly 2 cm in a few regions (Fig. A1c at $14°$, A1d at 16°N) but they are generally 1cm or less. The directional character of the baroclinic

waves and relatively small size of the Caribbean makes it challenging to use along-track filtering to isolate the barotropic and baroclinic components of tidal sea level. The second pertains to the model; namely, there are apparently no gross differences between the observed and modeled tide, at least for $M_2$ (shown) and other large tides which have been examined ($K_1$, and $O_1$; not shown). However, the tidal amplitudes are small enough, that even a 1 cm difference in amplitude is a significant fraction of the total amplitude.

*Author contributions.*   The author conceived and performed the research described in this manuscript using publicly-available data.

*Competing interests.*   The author has no competing interests in relation to this publication.

*Acknowledgements.*   Three anonymous reviewers and the editor are acknowledged for their contributions to improving this manuscript. AMSEAS model products are created by NAVOCEANO and distributed in cooperation with the Northern Gulf Institute; they are available at http://edac-dap.northerngulfinstitute.org/. Satellite altimeter data were extracted from the Radar Altimetry Database System (RADS),
available at http://rads.tudelft.nl/rads/rads.shtml. This work was supported by NASA awards NNX17AJ35G and NNX16AH88G.

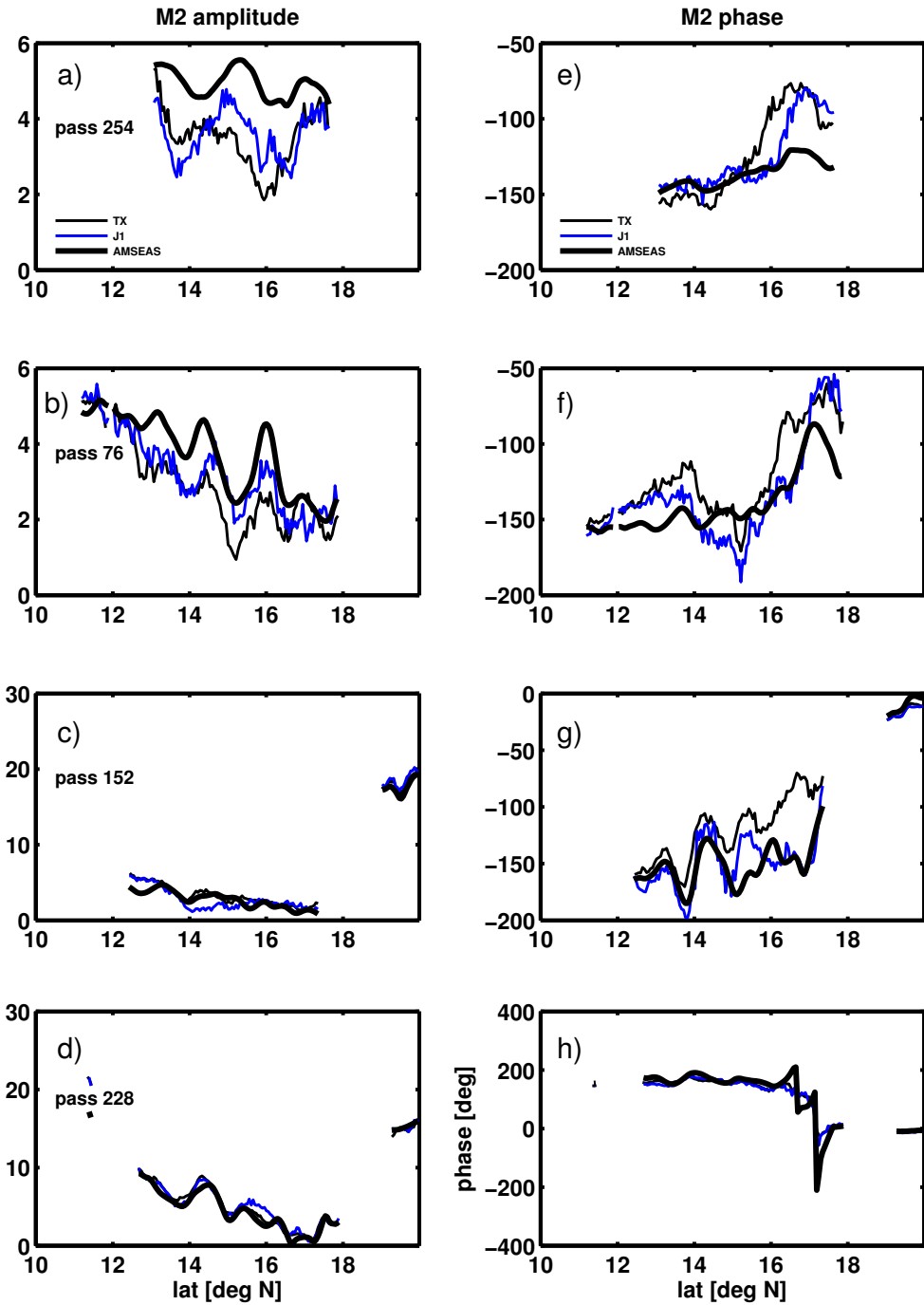

**Figure A1.** Comparison of satellite-derived and AMSEAS phase-locked tides in the Caribbean Sea: descending tracks.

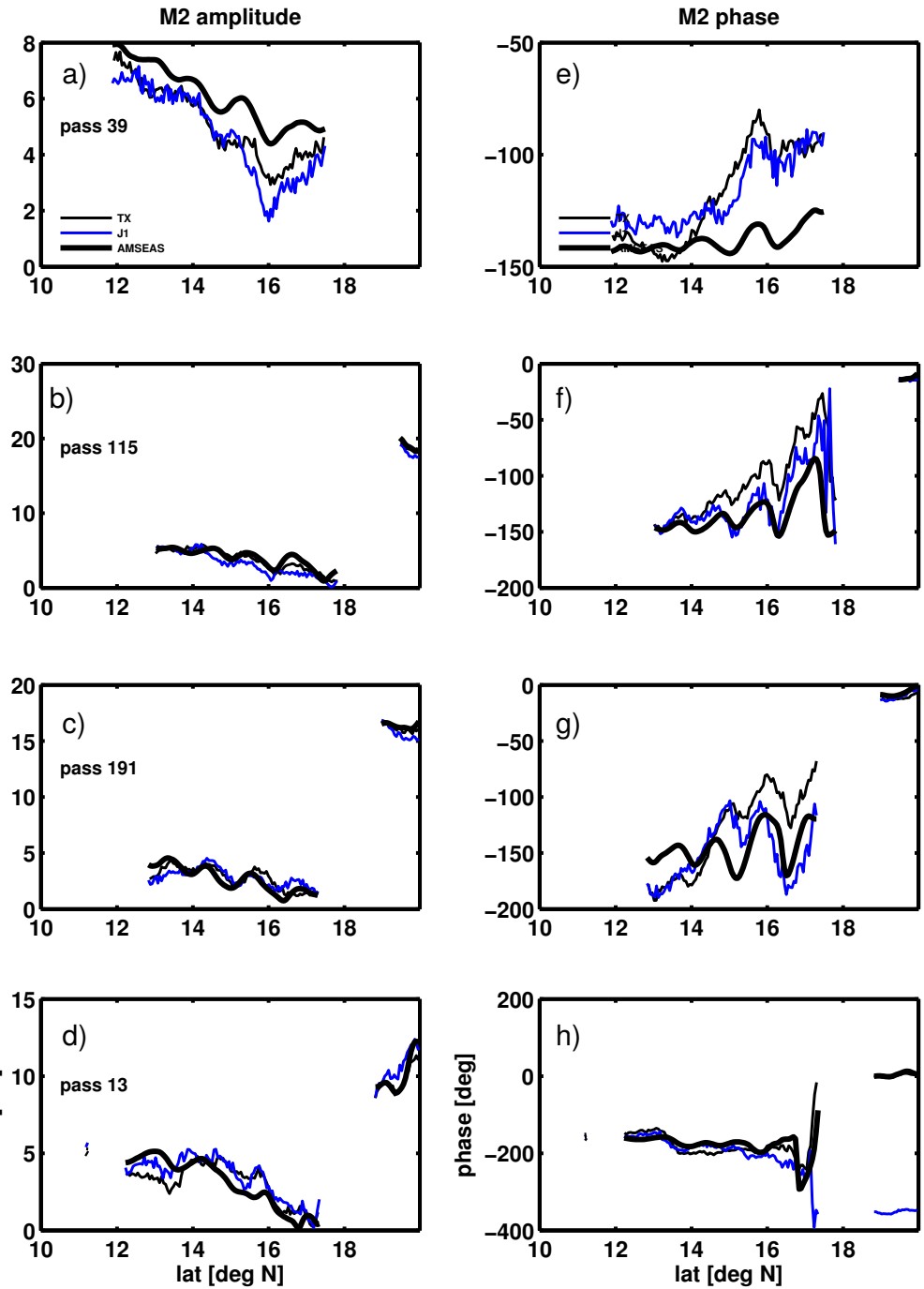

**Figure A2.** Comparison of satellite-derived and AMSEAS phase-locked tides in the Caribbean Sea: ascending tracks.

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
