# Peer review of "Predictability of Non-Phase-Locked Baroclinic Tides in the Caribbean Sea"

_Ocean Science, 2019_

## Referee Comment (RC1) · Anonymous Referee #1 · 29 May 2019

This manuscript describes the decomposition of low-frequency, tidally phase-locked and high frequency (assumed to be dominated by non-phase-locked) baroclinic sea surface height in an operational assimilated high-resolution ocean model of the Caribbean Sea (AMSEAS). The results are of regional interest, as there are few in-situ observations of internal tides in the Caribbean, but I feel that further analysis would give the paper wider appeal to the global internal wave/tide community. That said, I am not in the position to comment on the implications for the operational ocean model community. My suggestions for possible further work are to investigate interference between the multiple internal tide generation sites that are mentioned and/or the radial dispersion from Aves Escarpment.

Specific comments:

[Figure]

Page 1, Lines 20-22. Briefly explain how is is possible to separate the phase-locked baroclinic tide from the barotropic tide in satellite altimetry datasets.

Page 3, Line 12. ...sea surface heights data.

Figure 2. It is not clear to me that Figure 2 is necessary. You have already stated that the AMSEAS has been skill assessed and validated. I suggest Figure 2 and the related main text is cut or put in an appendix under a title of Further model validation (or similar).

Figure 3. This figure needs a colorbar.

Section 3. To confirm, both total (barotropic and baroclinic) sea level anomaly and baroclinic (steric, from temperature and salinity) sea level anomaly are output by the model? This needs to be clarified.

Section 3. How is it possible to separately resolve the M2 and S2 tides from a 96 hour timeseries? If I have missed something it need explaining more clearly.

Page 7, Lines 11-13. See also Guihou et al. (2017, JGR) and Aslam et al. (2018, Progress in Oceanography).

Section 4. The first two paragraphs of this section would be easier to understand if you referred to the example in Figure 6 earlier and used your standard notation (low frequency, phase-locked, and non-phase-locked) consistently. Non-phase-locked and high-frequency are used interchangeably despite stating on Page 9, Line 8 that the former will be used.

Figure 8. This figure would be improved if it had the same layout as Figure 6. RMS low frequency is stated as not shown, but it would be if the same figure layout was used.

Figure 9. Change forecast on the y-axis of panel a to fcst.

Figure 10. This figure would be improved if the line colours were used consistently, e.g., in panel a LF fcst is red so in panel b HF fcst should be red as well. Why is total

[Figure]

SLA not shown in panel b?

---

## Referee Comment (RC2) · Anonymous Referee #2 · 3 Jul 2019

This study examined the predictability of non-phase-locked internal tides in Caribbean sea using AMSES's model forecast. It is identified that the forecast error of the non-phase-locked tides is correlated with the forecast error of the sub-tidal (mesoscale) SLA. The forecast errors are spatially and temporally varying related to dynamical origin of the non-phase-locked tides.

The study is timely for the upcoming SWOT mission, to which the non-phase-locked tides and internal gravity waves can be a challenge in further scientific application. Understanding and predicting non-phase-locked tides can be valuable in utilizing the small-scale (15-150km) SSH signals.

The paper is well written and the analyses are well presented. I recommend publishing after a minor revision.

[Figure]

Main comments:

1. The non-phased-locked tides are defined as (page 8 line 5) the residual of the detided steric height minus its daily average. It contains both non-phased-locked tides and the supertidal internal gravity continuum. It may well be true that the non-phase-locked tides dominate the signal, but a frequency spectrum and variance quantification will be very useful for a proper justification, especially for the title, which can otherwise be misleading.

2. The model implementation of the tidal simulation needs elaboration.

Page 3 Line 10 reads: "Tides are not provided by the global model presently used for boundary conditions; instead, tides predicted with the OTIS barotropic tide model (Egbert and Erofeeva, 2002) are added to the barotropic currents and sea surface height data at open boundaries. In addition, the tide-generating force is applied within the model domain, and incorporates the effects of ocean loading and self-attraction consistent with the OTIS model.". But Page 6 line 8 reads "...have been subjected to least-squares harmonic analysis to identify the phase-locked tides at the M2, S2, K2, N2, 2N2, K1, O1, P1, and Q1 frequencies (provided through tidal open boundary conditions)". I may have missed something, but it could be helpful if more details were provided.

3. The phase-locked tides in the forecast need more validation.

3.1 Section 3 is dedicated to this evaluation, but needs more quantitative analyses and illustration. For example, the quantitative analyses in Appendix A are important and can be visualized in a map as a panel in figure 3.

3.2 As stated in the paper, the tide gauge cannot distinguish barotropic and baroclinic components, the mismatch between model forecast and tide gauge then cannot confirm the performance of the model forecast on phase-locked baroclinic tides.

3.3 The comparison in Figure 2 can be improved. It is said (Page 4 line 12) that

"The small offset between the observed and modeled spectra may be explained by the different time periods used for computing the spectra, which is the 2010–2012 period for AMSEAS and the 2002–2009 period for Jason-1." Why not compare the simultaneous forecast and the altimetry observations?

3.4 Another related question is "If large discrepancies exist in stationary tides between forecast and real ocean, will these discrepancies be contribute to errors in non-phase-locked tides?"

Minor comments:

Gaultier et al., 2016 may not be an ideal reference for the topic of "distinguishing balanced motion from inertia-gravity waves in sea surface topography data". Their regional ROMS model does not have tides.

---

## Referee Comment (RC3) · Anonymous Referee #3 · 9 Jul 2019

An interesting and useful study, showing variability in the sea level associated with phase-locked and non-phase-locked internal tides in the Caribbean Sea. In addition to providing nice results on the energetics of these waves in the model and satellite altimetry data, the paper describes methodology that can used for other regions and models. Presentation is very clear.

Specific remarks:

p. 16, line 16: "The findings of this study indicate that a substantial fraction of non-phase-locked tidal sea level variability may be predictable with an ocean forecast system"... Using self-verifying analyses, this is an overstatement. I guess I am picking on the use of word "predictable" here. As the author point out themselves, the forecast is verified against the nowcast, not independent data. The data assimilated in the now-

cast do not constrain the internal tide signal directly, and only potentially improve the background conditions for the internal tide propagation. We do not know if they do. All we can say from this analysis is that the internal wave propagation is sensitive to the changes in the background conditions implied by data assimilation. It is not shown (by comparison to the independent data) that nowcasts have better representation of the internal tide field.

The rationale for the use of the steric height is explained on p. 8, line 6. Can this note be moved to a place earlier in the text, before the first result using the steric height anomaly is discussed?

Can the author present details of how the steric height anomaly is computed? Steric height is discussed in textbooks as associated with thermal expansion of the water column, right? And NCOM is probably a Boussinesq model, conserving volume (the model water column does not expand due to the heating). Is there contradiction here? If the answer is common place, please ignore this remark. Otherwise, I would appreciate a short note in the text.

How is the phase-locked baroclinic tide separated from the barotropic tide along the track? The answer is probably in the earlier papers on the subject. Can the author provide a comment in this paper? Or point to a reference?

It is surprising to see the spatial structure of the non-phase-locked tide (Fig 6c) and its error (6d) look like radiating waves, similar to the phase-locked tide. If the eddies scatter the internal tide wave, resulting in the non-phase-locked tide, can one expect a more irregular pattern? In connection to this, what is the main difference in the subsurface background stratification of the T+0 and T+3.5 forecasts? Is it dominated by the eddy composition or more by the basin scale change in the vertical stratification (the depth and the strength of the thermocline)? As a suggestion, can the author show the detided, daily averaged zonal vertical sections of T and S (and/or density) in the top 500 m or so, at 14N say, T+3.5 and T+0 solutions, and their difference, for the same

[Figure]

date as Fig 6 and 7?

Fig 8: Can you discuss the possible reasons for the maximum rms amplitude of the non-phase-locked tide over Aves Escarpment? Can "Internal‐tide generation and destruction by shoaling internal tides" (Kelly and Nash 2010) be a possible mechanism?

Figure 11: the horizontal scale of the waves in the packets seems to be finer than the AMSEAS resolution. Again, the statement that "waves are predictable by the AMSEAS system" (in conclusions) is not supported by the analysis (e.g., of wave speed, dispersion properties, vertical structure, term balances, etc.).

Minor remarks:

Figure 3. Include the color bar.

p. 5, line 1: typo... "snapshot"

p. 9, line 27: typo ... "magnitude"

Figure 7: add the date in the caption.

p. 13, line 7: check the use of word "both"... seems to be out of place

---

## Editor Comment (EC1) · Philip Woodworth (Editor) · 18 Aug 2019

18 August 2019

Editor comments on "Predictability of Non-Phase-Locked Baroclinic Tides in the Caribbean Sea" by Edward D. Zaron

This area of modelling in tidal science is not my thing at all. However, it seems to me that the three reviewers made a number of detailed suggestions which the author has mostly taken on board for a revised version of the paper. I was not clear what R1 meant by 'further analysis' - if he means geographically that is obviously outside the scope of the paper.

I have only a few extra comments below and I look forward to seeing the revised ver-

[Figure]

sion.

Philip Woodworth

abstract line 2 - AMSEAS acronym perhaps needs defining

line 5 ... sea level anomalies (SLAs) ..

figure 1 caption. what does 'arbitrary representative date' mean? This sounds very jargonish. Also I take it that 2013-02-01 means 1 Feb, but dates expressed like this are always ambiguous. Could you go through the paper and check you make it clear where you have dates? Perhaps have a footnote upfront to say what date format is being used.

5 - certainly define AMSEAS here

figure 3 has different lat/lon annotation and style to figures 1 and 4 and uses east and not west longitude. And has no longitude (deg E) x/y caption etc.

4 - is Torres and Tsimplis 2012 an appropriate reference here? That is about the seasonal cycle. Maybe their 2011 paper?

p5, 4 - .. internal gravity waves and, specifically, the baroclinic tide.

p6, 10 - why five?

16 - southern coast –> South American coast

p7, 1-2 - why?

Figure 6 - These plots are all for 2013-02-01 again? Add to caption.

p9, 16 - some readers will now know what the T convention is for date/time, and in fact T does not tell you anything. It would be better to spell it out a bit more e.g. 1200 hours (UT) on 2013-02-01 or whatever.

Figure 7 - ditto as for figure 6

Figure 9 has fcst. in lower plot and forecast in upper

p15, 3 - sieches –> seiches

Also a more recent reference for the Puerto Rico seiches could be (not essential if you don't want): Woodworth, P.L. 2017. Seiches in the eastern Caribbean. Pure and Applied Geophysics, 174(12), 4283-4312, doi:10.1007/s00024-017-1715-7.

p17, Appendix A title. 'predicted' is the same as 'modelled', isn't it? You have predicted in the title and text on this page and 'mod' in Table A1.

phase should be phase lag

line 9 would better read The amplitude and Greenwich phase lag of the observed and modelled [or predicted] AMSEAS sea surface heights are listed.

I was a bit disappointed by the time I got to Table A1. These M2 values are all simply lifted from data centres and papers when there is a vast amount of new Caribbean data in, for example, the IOC sea level station monitoring facility.

---

## Author Response (AR1)

**Reply to Referees, OS-2019-53**
**Predictability of Non-Phase-Locked Baroclinic Tides in the Caribbean Sea**

Edward D. Zaron

August 30, 2019

Dear Editor:

This "Reply to Referees" addresses the comments from three anonymous referees and the editor. The original comments are reproduced, in Section 1, with each comment followed by my short reply, formatted in blue text. Line numbers mentioned here refer to the original version of the manuscript, except where noted. In Section 2 a summary of the comments is provided. Finally, in Section 3 the changes to the manuscript are summarized.

**1 Comments from Referees**

**Anonymous Referee #1 ()**

This manuscript describes the decomposition of low-frequency, tidally phase-locked and high frequency (assumed to be dominated by non-phase-locked) baroclinic sea surface height in an operational assimilated high-resolution ocean model of the Caribbean Sea (AMSEAS). The results are of regional interest, as there are few in-situ observations of internal tides in the Caribbean, but I feel that further analysis would give the paper wider appeal to the global internal wave/tide community. That said, I am not in the position to comment on the implications for the operational ocean model community. My suggestions for possible further work are to investigate interference between the multiple internal tide generation sites that are mentioned and/or the radial dispersion from Aves Escarpment.

While I understand the referee's sentiment that further analysis might broaden the appeal of this article, I do not think that AMSEAS is a good tool for such a study, especially for tides, because of the 96 hr forecast cycle. The short forecasts place constraints on the possibilities for dynamical studies with this model. Instead, I think it is best to investigate a question, e.g., regarding the predictability of non-phase-locked tides, which AMSEAS is uniquely suited to answer.

Specific comments:

Page 1, Lines 20-22. Briefly explain how is is possible to separate the phase-locked baroclinic tide from the barotropic tide in satellite altimetry datasets.

Done. Because the (low-mode) baroclinic waves closely follow the linear dispersion relation, their wavelength is much smaller than that of the barotropic waves. Separating the waves in altimetry

data is thus primarily an exercise in spatial filtering, which is a challenge due to the large inter-track separation, especially near the coastline or complex topography. Added at line 24: "In fact, it is only possible to separate the baroclinic tide from the barotropic tide in altimetry data because of the large separation in spatial scales between these classes of waves (Zaron 2019)."

Page 3, Line 12. ... sea surface heights data.

Done; "data" is plural and "heights" provides a parallel construction with "currents." I am undecided if this is a grammatical necessity, though, since it seems common to use "height" to refer to the plural "heights" of extended objects, e.g., "the sea surface height in the bay ...".

Figure 2. It is not clear to me that Figure 2 is necessary. You have already stated that the AMSEAS has been skill assessed and validated. I suggest Figure 2 and the related main text is cut or put in an appendix under a title of Further model validation (or similar).

Agreed. The Figure has been removed.

Figure 3. This figure needs a colorbar.

Done.

Section 3. To confirm, both total (barotropic and baroclinic) sea level anomaly and baroclinic (steric, from temperature and salinity) sea level anomaly are output by the model? This needs to be clarified.

No. The total sea level anomaly is output. I compute the baroclinic (steric) component from the temperature and salinity. I have re-organized and expanded Section 3 to clarify, as well as adding mention of this early in the manuscript, in the Introduction.

Section 3. How is it possible to separately resolve the M2 and S2 tides from a 96 hour timeseries? If I have missed something it need explaining more clearly.

The referee is correct. It is not possible to resolve the M2 and S2 tides from a single 96-hour time series, therefore, I estimate these tides from the entire two-year time series. I have slightly expanded the discussion in Section 3 to clarify how the phase-locked tides are estimated.

Page 7, Lines 11-13. See also Guihou et al. (2017, JGR) and Aslam et al. (2018, Progress in Oceanography).

I appreciate the pointer to these articles, which I was not previously aware of. They are now cited in the context of my remarks about the need for high model resolution.

Section 4. The first two paragraphs of this section would be easier to understand if you referred to the example in Figure 6 earlier and used your standard notation (low frequency, phase-locked, and non-phase-locked) consistently. Non-phase-locked and high-frequency are used interchangeably despite stating on Page 9, Line 8 that the former will be used.

I have revised this section to refer to Figure 6 earlier, and I no longer state that the non-phase-locked tide will be used instead of high-frequency.

Figure 8. This figure would be improved if it had the same layout as Figure 6. RMS low frequency is stated as not shown, but it would be if the same figure layout was used.

Agreed. Done.

Figure 9. Change forecast on the y-axis of panel a to fcst.

Done.

Figure 10. This figure would be improved if the line colours were used consistently, e.g., in panel a LF fcst is red so in panel b HF fcst should be red as well. Why is total SLA not shown in panel b?

I have revised the figure by using the line colors more consistently, and by putting the total steric height spectrum in both panels. Also, note that the spectra shown have been changed. The spectra

shown previously were *azimuthally integrated spectra* (i.e., the two-dimensional wavenumber spectra were integrated azimuthally with respect to wavenumber direction). The spectra now shown are so-called *isotropic spectra*; these are equal to $(2\pi k)^{-1}$ times the azimuthally integrated spectra. For isotropic random fields, the isotropic spectrum is the same as the one-dimensional spectrum, and this is the form of the spectrum which has normally been presented in studies of the mesoscale SSH spectrum (e.g., Callies and Wu, "Some expectations for submesoscale sea surface height variance spectra," to appear in *JPO, 2019*).

**Anonymous Referee #2 ()**

This study examined the predictability of non-phase-locked internal tides in the Caribbean Sea using AMSEAS's model forecast. It is identified that the forecast error of the non-phase-locked tides is correlated with the forecast error of the sub-tidal (mesoscale) SLA. The forecast errors are spatially and temporally varying related to dynamical origin of the non-phase-locked tides.

The study is timely for the upcoming SWOT mission, to which the non-phase-locked tides and internal gravity waves can be a challenge in further scientific application. Understanding and predicting non-phase-locked tides can be valuable in utilizing the small-scale (15-150km) SSH signals.

The paper is well written and the analyses are well presented. I recommend publishing after a minor revision.

Main comments:

1. The non-phased-locked tides are defined as (page 8 line 5) the residual of the detided steric height minus its daily average. It contains both non-phased-locked tides and the supertidal internal gravity continuum. It may well be true that the non-phase-locked tides dominate the signal, but a frequency spectrum and variance quantification will be very useful for a proper justification, especially for the title, which can otherwise be misleading.

Because the temporal frequency of the output, 3-hr, and the total duration of each model run, 96-hr, are only separated by a factor of 33, there are very few independent Fourier frequencies with which to build an estimate of the frequency spectrum. Longer time series formed by concatenating adjacent non-overlapping forecasts are also problematic because of discontinuities due to data assimilation. While I agree with the referee that there is interesting information in the frequency spectrum of model output, to analyze it in the present circumstance is not particularly useful. I do not understand how the title could be misleading in relation to the referee's comments.

2. The model implementation of the tidal simulation needs elaboration.

Page 3 Line 10 reads: "Tides are not provided by the global model presently used for boundary conditions; instead, tides predicted with the OTIS barotropic tide model (Egbert and Erofeeva, 2002) are added to the barotropic currents and sea surface height data at open boundaries. In addition, the tide-generating force is applied within the model domain, and incorporates the effects of ocean loading and self-attraction consistent with the OTIS model.". But Page 6 line 8 reads "... have been subjected to least-squares harmonic analysis to identify the phase-locked tides at the M2, S2, K2, N2, 2N2, K1, O1, P1, and Q1 frequencies (provided through tidal open boundary conditions)". I may have missed something, but it could be helpful if more details were provided.

I have reorganized this portion of the text, which may make this clearer. I changed the sentence structure and added a sentence on page 3 to explain that tides are not a part of HYCOM, so they

must instead be incorporated from another source, OTIS, on open boundaries. I have also added the phrase "and the tide-generating force", to the parenthetical note on page 6.

3. The phase-locked tides in the forecast need more validation.

I have expanded the appendix to inlude a graphical comparison to go with Section 3.

3.1 Section 3 is dedicated to this evaluation, but needs more quantitative analyses and illustration. For example, the quantitative analyses in Appendix A are important and can be visualized in a map as a panel in figure 3.

I do not see how to meaningfully represent the quantitative information in Appendix A in graphical form on Figure 3 (Figure 2 in the revised manuscript).

3.2 As stated in the paper, the tide gauge cannot distinguish barotropic and baroclinic components, the mismatch between model forecast and tide gauge then cannot confirm the performance of the model forecast on phase-locked baroclinic tides.

Precisely because of this point, I do not think it is useful to dwell on the validation of the phase-locked tides at tide gauges in the main text.

3.3 The comparison in Figure 2 can be improved. It is said (Page 4 line 12) that "The small offset between the observed and modeled spectra may be explained by the different time periods used for computing the spectra, which is the 2010–2012 period for AMSEAS and the 2002–2009 period for Jason-1." Why not compare the simultaneous forecast and the altimetry observations?

Following the suggestion of Referee #1, this figure has been omitted.

Note, though, it is not straightforward to meaningfully compare the model and observed SSH spectra over the same time period. The altimeter sampling time is about 10 days, so each altimeter track only contributes 36 degrees of freedom to the spectral estimate in a year. In contrast, the model output is every 3 hours, and there are up to 5 model estimates for any given date-time. Thus, the spectral estimate from two years of model output contains many more degrees of freedom than an altimeter-derived estimate from two years.

3.4 Another related question is "If large discrepancies exist in stationary tides between forecast and real ocean, will these discrepancies be contribute to errors in non-phase- locked tides?"

Agreed. I have expanded two paragraphs to highlight this question. First, in Section 3 on the phase-locked tides, I mention that the AMSEAS model has not been calibrated w.r.t. the baroclinic tides and I point to a newly-added graphical model-data comparison in the Appendix. Second, in the final paragraph of Section 6, the Discussion, I explicitly mention that error in the phase-locked tide is a type of systematic error which cannot be detected in the self-verifying analysis.

Minor comments:

Gaultier et al., 2016 may not be an ideal reference for the topic of "distinguishing bal- anced motion from inertia-gravity waves in sea surface topography data". Their regional ROMS model does not have tides.

This reference was intended to provide a pointer to more information about SWOT, rather than inertia-gravity waves per se. It is now omitted.

**Anonymous Referee #3 ()**

An interesting and useful study, showing variability in the sea level associated with phase-locked and non-phase-locked internal tides in the Caribbean Sea. In addition to providing nice results on the energetics of these waves in the model and satellite altimetry data, the paper describes methodology that can used for other regions and models. Presentation is very clear.

Specific remarks:

p. 16, line 16: "The findings of this study indicate that a substantial fraction of non-phase-locked tidal sea level variability may be predictable with an ocean forecast system" ... Using self-verifying analyses, this is an overstatement. I guess I am picking on the use of word "predictable" here. As the author point out themselves, the forecast is verified against the nowcast, not independent data. The data assimilated in the now-cast do not constrain the internal tide signal directly, and only potentially improve the background conditions for the internal tide propagation. We do not know if they do. All we can say from this analysis is that the internal wave propagation is sensitive to the changes in the background conditions implied by data assimilation. It is not shown (by comparison to the independent data) that nowcasts have better representation of the internal tide field.

Agreed. I believe that the explanation of the self-verifying analysis is clear enough to avoid confusion, though. I changed the third paragraph of the Introduction to explain the rationale for using self-verifying analyses earlier in the paper.

The rationale for the use of the steric height is explained on p. 8, line 6. Can this note be moved to a place earlier in the text, before the first result using the steric height anomaly is discussed?

Yes, the rationale for the use of steric height has been moved to the third paragraph of the Introduction.

Can the author present details of how the steric height anomaly is computed? Steric height is discussed in textbooks as associated with thermal expansion of the water column, right? And NCOM is probably a Boussinesq model, conserving volume (the model water column does not expand due to the heating). Is there contradiction here? If the answer is common place, please ignore this remark. Otherwise, I would appreciate a short note in the text.

An explanation of steric height anomaly has been expanded to address the questions of the Referees. I have also included, below, plots of the sea level anomaly ($\Delta$SLA) and the sea level anomaly minus the steric height anomaly ($\Delta$(SLA-SHA)) in order to exhibit the degree to which the steric height anomaly (which is computed from the vertical profiles of the model's three-dimensional temperature and salinity fields) compensates the apparent steric height anomaly in the output SLA. The plots correspond to anomalies computed as the difference between two arbitrary dates 6 hours apart, 2019-01-04T18:00:00 and 2019-01-05T00:00:00, to visually emphasize the dominant signal, which is the semidiurnal barotropic tide. Please see the caption for more information.

How is the phase-locked baroclinic tide separated from the barotropic tide along the track? The answer is probably in the earlier papers on the subject. Can the author provide a comment in this paper? Or point to a reference?

Essentially, the only way to separate baroclinic and barotropic tide sea level from along track data is via spatial filtering, either through simple one-dimensional along-track filtering or through more complex filtering which uses multiple tracks to map the two-dimensional wave field. I added a reference to Zaron 2019 and short explanation at the bottom of page 1 in the revision.

It is surprising to see the spatial structure of the non-phase-locked tide (Fig 6c) and its error (6d) look like radiating waves, similar to the phase-locked tide. If the eddies scatter the internal tide wave, resulting in the non-phase-locked tide, can one expect a more irregular pattern? In connection to this, what is the main difference in the subsurface background stratification of the T+0 and T+3.5 forecasts? Is it dominated by the eddy composition or more by the basin scale change in the vertical stratification (the depth and the strength of the thermocline)? As a suggestion, can the author show the detided, daily averaged zonal vertical sections of T and S (and/or density) in the top 500 m or so, at 14N say, T+3.5 and T+0 solutions, and their difference, for the same date

as Fig 6 and 7?

A figure has been added to the manuscript (new Figure 7), together with a descriptive paragraph, following the suggestion of the referee.

Fig 8: Can you discuss the possible reasons for the maximum rms amplitude of the non-phase-locked tide over Aves Escarpment? Can "Internal tide generation and destruction by shoaling internal tides" (Kelly and Nash 2010) be a possible mechanism?

A discussion has been added in the context of the figure mentioned above. In essence, the maximum appears to be related to the appearance of sawtooth-shaped waveforms, apparently indicative of mode-1 nonlinear wave steepening at sites on Aves Escarpment. Small errors in the position of the leading edge of these waveforms leads to large errors.

Figure 11: the horizontal scale of the waves in the packets seems to be finer than the AMSEAS resolution. Again, the statement that "waves are predictable by the AMSEAS system" (in conclusions) is not supported by the analysis (e.g., of wave speed, dispersion properties, vertical structure, term balances, etc.).

The referee seems to misunderstand which part of the text is referring to Figure 11. I assume the referee is referring to the sentence on p. 15, line 4, in the Discussion (not in the Conclusions), "Figure 10b indicates that a considerable fraction of the variance associated with these waves is predictable by the AMSEAS system". The antecedent of "these waves" was apparently not clear enough, but this sentence refers explicitly to the results in Figure 10b, not Figure 11, and the data in the figure definitely support this statement. In order to make the antecedent explicit, I have replaced "these waves" with "the wavenumber peak near 60 km." I have also added a sentence at the end of the paragraph to note the referee's point that the waves in the sunglint images are not what is being represented in the AMSEAS output.

Minor remarks:

Figure 3. Include the color bar.

Done.

p. 5, line 1: typo ... "snapshot"

Done.

p. 9, line 27: typo ... "magnitude"

Done.

Figure 7: add the date in the caption.

Done.

p. 13, line 7: check the use of word "both" ... seems to be out of place

Omitted. Meaning is unchanged.

**Editor Comments:**

18 August 2019 Editor comments on "Predictability of Non-Phase-Locked Baroclinic Tides in the Caribbean Sea" by Edward D. Zaron This area of modelling in tidal science is not my thing at all. However, it seems to me that the three reviewers made a number of detailed suggestions which the author has mostly taken on board for a revised version of the paper. I was not clear what R1 meant by 'further analysis' – if he means geographically that is obviously outside the scope of the paper. I have only a few extra comments below and I look forward to seeing the revised version.

Philip Woodworth

abstract line 2 - AMSEAS acronym perhaps needs defining

"AMSEAS" is not an acronym. I believe it is best to regard it as a proper noun with peculiar captialization.

line 5 ... sea level anomalies (SLAs) ..

Done.

figure 1 caption. What does 'arbitrary representative date' mean? This sounds very jargonish. Also I take it that 2013-02-01 means 1 Feb, but dates expressed like this are always ambiguous. Could you go through the paper and check you make it clear where you have dates? Perhaps have a footnote upfront to say what date format is being used.

I have omitted the word, "arbitrary," for clarity. I have added a parenthetical note to the figure caption to indicate that ISO 8601 date format is used throughout the manuscript.

5 - certainly define AMSEAS here

Added: "The name, "AMSEAS," is not an acronym, it is the captialized contraction of "American Seas" which has been adopted as the system's name."

figure 3 has different lat/lon annotation and style to figures 1 and 4 and uses east and not west longitude. And has no longitude (deg E) x/y caption etc.

The figure (now Figure 2) has been rendered in the same style as the other figures.

4 - is Torres and Tsimplis 2012 an appropriate reference here? That is about the seasonal cycle. Maybe their 2011 paper?

Thank you. That was an oversight.

p5, 4 - .. internal gravity waves and, specifically, the baroclinic tide.

Added "and."

p6, 10 - why five?

At hour $0Z$ of any day, there are 5 estimates (1 nowcast + 4 forecasts) for the ocean state.

16 - southern coast $\rightarrow$ South American coast

Done.

p7, 1-2 - why?

I assume the comment is in relation to the statement that a resolution of 1-2$\mathrm{km}$ is needed to accurately model the generation of low-mode baroclinic tides. Many factors influence the resolution requirements. In response to Referee #1 I have added additional references to this statement.

Figure 6 - These plots are all for 2013-02-01 again? Add to caption.

Done. Date added to caption.

p9, 16 - some readers will not know what the T convention is for date/time, and in fact T does not tell you anything. It would be better to spell it out a bit more e.g. 1200 hours (UT) on 2013-02-01 or whatever.

I have added a parethetical note which spells out the time at the first usage of the ISO 8601 date and time notation.

Figure 7 - ditto as for figure 6

Done.

Figure 9 has fcst. in lower plot and forecast in upper

Done.

p15, 3 - sieches $\rightarrow$ seiches Also a more recent reference for the Puerto Rico seiches could be (not essential if you don't want): Woodworth, P.L. 2017. Seiches in the eastern Caribbean. Pure and Applied Geophysics, 174(12), 4283-4312, doi:10.1007/s00024-017-1715-7.

Done. Thank you for the pointer to this article.

p17, Appendix A title. 'predicted' is the same as 'modelled', isn't it? You have predicted in the title and text on this page and 'mod' in Table A1. phase should be phase lag

Done. Changed to "predicted" throughout the Appendix.

line 9 would better read The amplitude and Greenwich phase lag of the observed and modelled [or predicted] AMSEAS sea surface heights are listed.

Done.

I was a bit disappointed by the time I got to Table A1. These M2 values are all simply lifted from data centres and papers when there is a vast amount of new Caribbean data in, for example, the IOC sea level station monitoring facility.

Thank you for this pointer to the IOC data. I will use the new data to conduct subsequent work in this area, but I believe the present comparison is adequate merely to demonstrate that AMSEAS is realistic enough that we can use it to learn about the predictability of the internal tide. Future study of the model accuracy per se, using independent data, should utilize the newer data.

**2 Author's Response**

Referee #1 suggests a number of very useful points for clarification, but his or her overall opinion seems to be that additional analysis should be provided in order to give the paper wider appeal or applicability. While I can see the merit in this suggestion, I believe the paper benefits from the present approach in which AMSEAS is used rather narrowly to assess the feasibility of predicting the sea-level signal of non-phase-locked tides. A more comprehensive study of interference of tides generated at multiple sites or the radial dispersion of the tide from Aves Escarpment, topics suggested by the referee, would be interesting and useful, but studying these phenomena or dynamics with AMSEAS would be unnecessarily difficult because of its operational 96-hour forecast cycle. Instead, I have tried to answer a rather narrow question with AMSEAS, which I believe it is uniquely suited to answer. The dynamical questions would be better addressed with long simulations from a high-resolution model, and I have not attempted to address them in the revised manuscript.

The comments of Referee #2 identify a number of useful areas for clarification, with the most significant point being that the phase-locked tides in the forecast need more validation. I agree that this is useful, but the referee's comments suggest why this is challenging, namely: (1) tide gauges cannot distinguish barotropic and baroclinic contributions to sea level, and (2) direct comparisons of altimetry and model output in the time domain are of limited use because the once-per-ten-day sampling (or less) of the altimeters leads to many fewer degrees of freedom in an altimeter-based quantity, compared to a model-derived quantity.

In partial contradiction to Referee #2, Referee #1 feels that Figure 2 is not necessary since a more meaningful skill assessment and validation of AMSEAS has already appeared elsewhere.

Referee #3 makes the important point that my usage of the phrase "prediction error" may be misinterpreted since the metrics reported in the paper mostly concern self-verifying analysis. In addition to this over-arching comment, Referee #3 also requests more investigation of the causes of the non-phase-locked tide, especially the evolution of the subsurface stratification.

I have not had time to investigate, in detail, the estimation of the prediction error using independent data with AMSEAS. There are several challenges which make it difficult to compute a meaningful estimate of this quantity. The key issue is that the model output is only available at 3-hour intervals, but the predictions must be compared at the same time as the altimeter measurements, which do not exactly coincide with the model output. It is relatively straightforward to envision how to compare the low-frequency variability of the model with observations (and this has been done in the validation studies cited in the text), but a comparison of the high-frequency component is ambiguous. The low-frequency signal could be defined at the 3-hourly output times by linear interpolation of the daily average low-frequency fields (defined herein at $T + 0.5$, ..., $T + 3.5$); and a high-frequency signal could be computed as the residual of the detided steric height at the same 3-hourly times. However, how one would interpolate the high-frequency signal from the 3-hourly times to the observation times is unclear, and the tidal variability within the 3-hour intervals would be significant. While I think it would be a useful and necessary effort to compute the prediction error using independent data, I believe it should be the subject of a dedicated study, and it would not integrate well with the present manuscript.

**3   Changes in the Manuscript**

Detailed changes in the manuscript have been enumerated under the referees' comments above, in Section 1.

Changes in the manuscript have been made in response to the referees' queries for more explanation regarding the computation of steric height, especially to clarify that this is a derived quantity, not part of the native output of AMSEAS. The third paragraph in the Introduction now explains up front the rationale for analysis of steric height rather than total sea level. Then, in Section 3, the details of the steric height calculation are presented, and some references to the literature are provided to indicate caveats connected with using this quantity.

I have made some additional changes to Section 3 to address the referees' concern with validating the phase-locked tides in the model. I now refer to the Appendix where graphical comparisons with harmonic constants estimated from altimetry are shown. This section is non-quantitative, and it is intended to illustrate the characteristics of possible errors in the phase-locked tides. It highlights the eventual need for calibrating the phase-locked tides, but it points out two factors which will make this a challenge in practice: (1) there are significant uncertainties in the baroclinic tides estimated from altimetry, so it is not presently clear what data source should be relied on for model calibration, and (2) the fundamental approach to calibrating a model is also not clear (Can it be accomplished by simply increasing model resolution and acquiring adequate bathymetry? Can the calibration of phase-locked tides be addressed independently of non-phase-locked tides when both are simultaneously resolved?) Although these are important issues, I think it is best not to delve into them in this paper.

Several significant changes have been made in the presentation of the Figures. The original Figure 2, which compared observed and model wavenumber spectra, has been omitted. A new panel was added to the original Figure 8 (now Figure 9) in order to match the layout of the original Figure 6. A new Figure 6 has been added which exhibits a zonal transect across $13.7°$N, and a paragraph has been added to describe the results in the figure. Lastly, the original Figure 10 has been revised by changing the line styles, and by re-scaling the spectra by $(2\pi k)^{-1}$ so that they are one-dimensional *isotropic* wavenumber spectra.

[Figure]

Figure 1: A demonstration of the degree to which the steric height anomaly (SHA; computed from AMSEAS model output) represents baroclinic effects in the sea level anomaly (SLA; native AMSEAS model output). The top panel shows the SLA difference between two arbitrary dates separated by 6 hours, 2019-01-04T18:00:00Z and 2019-01-05T00:00:00Z, to visually emphasize the dominant signal of the semidiurnal barotropic tide. Smaller-scale features related to the baroclinic tide and mesoscale are also evident. The bottom panel shows the SLA anomaly minus the SHA anomaly. It is apparent that the baroclinic features have been greatly reduced. Note that some small baroclinic features appear to remain; transects through the fields (not shown, but these have been computed for several arbitrary times) indicate that about 10% of the SLA baroclinc signal is not compensated by the SHA, based on a visual assessment. The reasons for this have not been investigated, but there are several possibilities: (1) the underlying NCOM model may use a different equation of state compared to the TEOS library which was used to compute the steric height, (2) spatial truncation errors related to the transformation of the AMSEAS hybrid (sigma-z) grid to the fixed output grid may contribute error, or (3) temporal truncation errors related to the barotropic-baroclinic time-splitting may be signficant.